# Structural mapping of oligomeric intermediates in an amyloid assembly pathway

Theodoros K Karamanos[1,2†], Matthew P Jackson[1,2], Antonio N Calabrese[1,2], Sophia C Goodchild[1,2‡], Emma E Cawood[1,2], Gary S Thompson[1,2§], Arnout P Kalverda[1,2], Eric W Hewitt[1,2], Sheena E Radford[1,2]*

[1]The Astbury Centre for Structural Molecular Biology, University of Leeds, Leeds, United Kingdom; [2]School of Molecular and Cellular Biology, University of Leeds, Leeds, United Kingdom

*For correspondence:
s.e.radford@leeds.ac.uk

Present address: †National Institute of Diabetes and Digestive and Kidney Diseases, National Institutes of Health, United States; ‡Department of Molecular Sciences, Macquarie University, Sydney, Australia; §School of Biosciences, University of Kent, Canterbury, United Kingdom

Competing interests: The authors declare that no competing interests exist.

**Abstract** Transient oligomers are commonly formed in the early stages of amyloid assembly. Determining the structure(s) of these species and defining their role(s) in assembly is key to devising new routes to control disease. Here, using a combination of chemical kinetics, NMR spectroscopy and other biophysical methods, we identify and structurally characterize the oligomers required for amyloid assembly of the protein $\Delta N6$, a truncation variant of human $\beta_2$-microglobulin ($\beta_2 m$) found in amyloid deposits in the joints of patients with dialysis-related amyloidosis. The results reveal an assembly pathway which is initiated by the formation of head-to-head non-toxic dimers and hexamers *en route* to amyloid fibrils. Comparison with inhibitory dimers shows that precise subunit organization determines amyloid assembly, while dynamics in the C-terminal strand hint to the initiation of cross-$\beta$ structure formation. The results provide a detailed structural view of early amyloid assembly involving structured species that are not cytotoxic.
DOI: https://doi.org/10.7554/eLife.46574.001

## Introduction

Oligomers have been the focus of amyloid research over decades because of their pivotal role in assembly and their potential cytotoxicity (*Chiti and Dobson, 2017*). Numerous aggregation-prone proteins (or their fragments) form oligomers (*Benilova et al., 2012*; *Wei et al., 2011*; *Laganowsky et al., 2012*; *Apostol et al., 2013*), some of which are cytotoxic (*Laganowsky et al., 2012*; *Ono et al., 2009*; *Fusco et al., 2017*), while others are not (*Mannini et al., 2014*). Many groups have attempted to elucidate the structure(s) of amyloid oligomers with different biological properties (*Chiti and Dobson, 2017*). However, their ephemeral nature, dynamic signature, and heterogeneity in mass and conformation provide significant experimental challenges. Hence, our current understanding of the structure of oligomers is often limited to low-resolution models (*Fusco et al., 2017*; *Vestergaard et al., 2007*; *Campioni et al., 2010*; *Cremades et al., 2012*), or to oligomers assembled from non-natural amino acids, short peptides, or protein fragments (*Laganowsky et al., 2012*; *Apostol et al., 2013*; *Sangwan et al., 2017*). Establishing a relationship between the oligomers observed and the mechanism of amyloid formation is also an important, but challenging, task. In some cases, oligomers have been shown to be 'off-pathway' since they have to dissociate for amyloid formation to proceed (*Wu et al., 2010*; *Baskakov et al., 2002*; *Souillac et al., 2003*; *Bieschke et al., 2010*). Characterization of such species, however, does not provide insight into the structural mechanism by which initially unstructured (e.g. Aβ42, α-synuclein) or natively structured proteins (e.g. lysozyme, transthyretin, antibody light chains, β₂-microglobulin (β₂m)) undergo conformational conversion to form the parallel in-register cross-β structure of amyloid

(*Iadanza et al., 2018a*). Other oligomers have been shown to be on-pathway (*Fusco et al., 2017*; *Cremades et al., 2012*), or to form via secondary nucleation processes that enhance the rate of fibril formation (*Cohen et al., 2013*). Proteins in an oligomeric or aggregated form have also been characterized kinetically, thermodynamically and biophysically (*Cohen et al., 2013*; *Cohen et al., 2018*; *Lenton et al., 2017*). However, a detailed understanding of *both* the structural properties of oligomers *and* their role in assembly is needed in order to understand the structural mechanism(s) of amyloid formation and the origins of cytotoxicity, as well as to design inhibitors of the assembly process.

Here, we describe an integrative approach which uses kinetic modeling to identify oligomers formed on-pathway to fibril formation, NMR spectroscopy and other biophysical methods to determine their structural properties, and cellular assays to determine their cytotoxicity. The strategy employed can be applied to other assembling protein systems and draws on the powers of NMR to provide detailed structural information about individual precursors in dynamic equilibrium within complex mixtures of assembling species, and kinetic modeling to ascribe their role in amyloid formation. To exemplify this combined structural and kinetic approach we focus on the naturally occurring variant of human $\beta_2$m, known as $\Delta$N6. This variant lacks the N-terminal six amino acids and is formed by natural proteolytic truncation of the wild-type (WT) protein (*Esposito et al., 2000*; *Eichner et al., 2011*). WT human $\beta_2$m (named herein as h$\beta_2$m) forms amyloid deposits in the joints of patients undergoing long term hemodialysis (*Gejyo et al., 1985*). However, h$\beta_2$m does not aggregate into amyloid fibrils at physiologically relevant pH and temperature on an experimentally accessible timescale in vitro (the pH in normal and diseased joints ranges from 5.5 to 7.4; *Floege and Ehlerding, 1996*). Addition of $Cu^{2+}$ ions, detergents, organic solvents, glycosaminoglycans or collagen can drive h$\beta_2$m amyloid formation at neutral pH (*Platt and Radford, 2009*; *Stoppini and Bellotti, 2015*; *Yamamoto et al., 2005*; *Benseny-Cases et al., 2019*). These reagents partially unfold the native protein, facilitating *cis-trans* isomerization of Pro32 that initiates assembly (*Eichner et al., 2011*; *Platt and Radford, 2009*; *Yamamoto et al., 2005*; *Chiti et al., 2001*). By contrast with the intransigence of h$\beta_2$m to form amyloid in vitro, $\Delta$N6 is highly amyloidogenic, forming fibrils rapidly in vitro in the absence of additives at pH 6–7 (*Karamanos et al., 2016*). $\Delta$N6 forms ~ 30% of $\beta_2$m in amyloid plaques in patients with dialysis-related amyloidosis (*Bellotti et al., 1998*). Previous studies have shown that $\Delta$N6 can induce amyloid formation of h$\beta_2$m at near-neutral pH in vitro (*Eichner et al., 2011*) and can co-assemble with the WT protein into amyloid fibrils (*Sarell et al., 2013*). The structure of h$\beta_2$m in amyloid fibrils formed in vitro at low pH (pH 2.0) has also been solved recently using solid-state NMR and cryo-EM, revealing a parallel in-register cross-$\beta$ structure typical of amyloid, which differs dramatically from the all anti-parallel immunoglobulin fold of the native precursor (*Iadanza et al., 2018b*). The atomic structure(s) of h$\beta_2$m amyloid fibrils formed in vivo, and those of $\Delta$N6 formed in vitro or ex vivo, however, are not yet known.

Several examples of oligomers (dimers, tetramers and hexamers) of WT h$\beta_2$m have been reported previously (*Calabrese et al., 2008*; *Eakin et al., 2006*; *Mendoza et al., 2011*; *Mendoza et al., 2010*; *Halabelian et al., 2015*; *Colombo et al., 2012*; *Rennella et al., 2013*; *Liu et al., 2011*; *Karamanos et al., 2014*), with one report of a domain swapped dimer of $\Delta$N6 stabilized by addition of a nanobody (*Domanska et al., 2011*). Since h$\beta_2$m is inert to aggregation at physiological pH and temperature in vitro, the oligomerization of the protein was stimulated by mutation and/or the addition of $Cu^{2+}$ ions (*Calabrese et al., 2008*; *Eakin et al., 2006*; *Mendoza et al., 2011*; *Mendoza et al., 2010*), or by linkage of monomers via non-native disulfide bonds (*Halabelian et al., 2015*; *Colombo et al., 2012*). Although some of these oligomers form under conditions in which WT h$\beta_2$m may eventually form fibrils, the role of individual oligomeric species in the aggregation mechanism remains unclear. The oligomers formed in the initiating stages of aggregation of $\Delta$N6 into amyloid also remain obscure.

Here, we show that amyloid formation of $\Delta$N6 occurs via a remarkably specific assembly mechanism involving the transient formation of dimers and hexamers. Exploiting NMR methods able to analyze dynamic and lowly populated states (*Anthis and Clore, 2015*), we characterize these assemblies, yielding a structural model of the initiating events in $\Delta$N6 aggregation in atomic detail. The results reveal the formation of head-to-head dimers that pack into symmetric hexamers that retain a native-like immunoglobulin fold and are not cytotoxic. The hexamers appear to be primed for further conformational change into the cross-$\beta$ structure of amyloid by dynamic unfurling of their C-terminal $\beta$-strands. The results portray a detailed atomic view of the early stages of $\Delta$N6 assembly that may

enable the development of routes to combat disease by targeting the specific protein-protein interactions that define the early stages of assembly.

## Results

### Fibril elongation occurs through an oligomeric state

Previous results have shown that ΔN6 assembles rapidly into amyloid fibrils in vitro at pH 6.2, but not at pH 8.2 (*Eichner et al., 2011*), suggesting that lowering the pH increases the population of aggregation-prone species. Such species may also be relevant in vivo given the acidic microenvironment of the joints of DRA patients (*Eichner et al., 2011*; *Bellotti et al., 1998*; *Karamanos et al., 2014*). At pH 6.2 (close to its pI of 5.8) ΔN6 is dynamic, but retains a native-like immunoglobulin fold (*Eichner et al., 2011*). To determine the kinetic mechanism by which ΔN6 aggregates into amyloid fibrils, experiments were performed in which ΔN6 fibril seeds (20 µM monomer equivalent concentration) were incubated with different concentrations of ΔN6 monomers (20 µM to 500 µM) and the rate of amyloid formation was monitored by the fluorescence of thioflavin T (ThT). All experiments were performed at pH 6.2 at a total ionic strength of 100 mM (see Materials and methods). The simplest kinetic mechanism in which monomers add to the fibril ends would result in a linear dependence of the initial rate of fibril elongation *versus* the monomer concentration, with saturation at high monomer concentrations (*Buell et al., 2014*; *Buell et al., 2010*; *Xue et al., 2009*). Such behavior is observed for seeded assembly of acid unfolded monomers of h$\beta_2$m, which initially lack persistent structure (*Platt et al., 2008*), into amyloid fibrils at pH 2.0 (*Figure 1a,b*). By contrast, ΔN6 showed more complex behavior, with a clear non-linearity in the initial rate of elongation versus monomer concentration, in which rapid seeded growth occurs only above ~200 µM ΔN6 (*Figure 1c, d*). This indicates that fibril elongation by ΔN6 must involve addition of one or more oligomeric species to the fibril ends under the conditions employed.

### Native-like dimers and hexamers form during Δn6 assembly

The concentration-dependence of ΔN6 elongation could be explained by an oligomer(s) acting as the elongation unit. To explore whether oligomeric species of ΔN6 are formed under the conditions employed, sedimentation velocity analytical ultracentrifugation (AUC), size exclusion chromatography (SEC), cross-linking, and NMR experiments were performed. These approaches report on the conformational properties and molecular weight distribution of the assemblies formed at different ΔN6 concentrations. Sedimentation velocity AUC experiments showed that ΔN6 forms discrete oligomers at pH 6.2, with monomers, dimers and higher order species with a sedimentation coefficient (S value) consistent with 6–9-mers (although the rapid equilibration of the species present prevents accurate determination of their mass and population) (*Figure 2a*). To investigate the molecular mass of the species present, ΔN6 was cross-linked after different incubation times in the absence of fibril seeds using 1-ethyl-3-(3-dimethylaminopropyl)-carbodiimide hydrochloride (EDC) (see Materials and methods) and the resulting species examined using SDS-PAGE (*Figure 2b*). This revealed the presence of hexamers during assembly (*Figure 2b*). The population of the hexameric species is decreased at later time points, presumably because it is consumed into fibrils (*Figure 2b*). Analytical SEC of ΔN6 at different protein concentrations without cross-linking revealed only monomers and dimers (*Figure 2—figure supplement 1a*), consistent with the higher order assemblies dissociating upon dilution on the column. However, when cross-linking was performed prior to SEC, higher molecular weight oligomers were observed, with these species being more abundant when higher protein concentrations were used (*Figure 2—figure supplement 1b,c*). At the highest concentration of ΔN6 used (500 µM) cross-linking resulted in the rapid formation of high molecular weight aggregates that elute in the void volume (*Figure 2—figure supplement 1b,c*). The population of these aggregates increases with time, accompanied by depletion of the oligomers, consistent with these species being capable of assembly into amyloid (*Figure 2—figure supplement 1d*).

[1]H-NMR and [1]H-[15]N HSQC NMR spectra of ΔN6 were next acquired to examine the properties of the oligomers formed. Significant changes in chemical shift and linewidth of individual resonances at different concentrations of ΔN6 were observed, consistent with the finding that ΔN6 self-assembles into higher molecular weight species at pH 6.2 (*Figure 2c* and *Figure 2—figure supplement 2a*). The residues most affected lie in the A strand and the BC, DE and FG loops, suggesting that

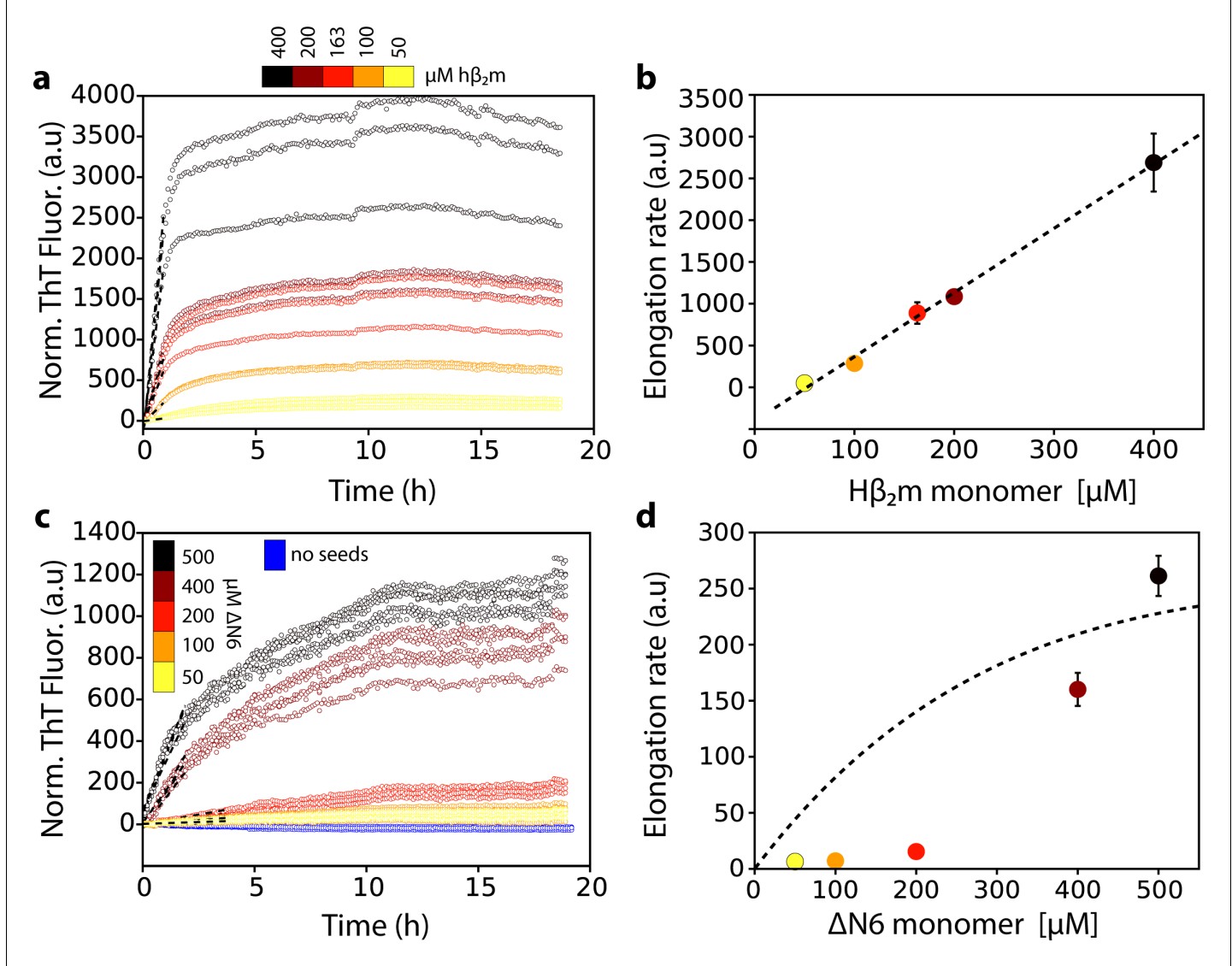

**Figure 1.** Dependence of the fibril elongation rate on the concentration of soluble protein. Seeded elongation assays for (**a**) h$\beta_2$m at pH 2.0 monitored by ThT fluorescence. 20 µM of preformed seeds of h$\beta_2$m (formed at pH 2.0) and varying amounts of soluble protein were added, as indicated in the key. Note that the protein does not aggregate under these conditions in the absence of seeds on this timescale (*Xue et al., 2008*). The dashed line shows the initial rate of each reaction. (**b**) The initial rate of fibril elongation (shown in units of ThT fluorescence (a.u.)/h) versus the concentration of h$\beta_2$m added. The dashed line represents a prediction using a monomer addition model (see *Table 4*). (**c**) Seeded elongation assays for $\Delta$N6 using 20 µM preformed seeds formed from $\Delta$N6 at pH 6.2 as a function of the concentration of soluble $\Delta$N6 added. Open blue symbols denote the ThT fluorescence signal of 500 µM $\Delta$N6 in the absence of seeds. The dashed line shows the initial rate of each reaction. (**d**) The initial rate of fibril elongation (shown in units of ThT fluorescence (a.u.)/h) versus the concentration of soluble $\Delta$N6 added. The dashed line shows the dependence of the elongation rate (in units of ThT fluorescence (a.u.)/h) on the concentration of monomer assuming a monomer addition model (see *Table 4*). The elongation rate for monomer addition shows a hyperbolic behavior as a function of monomer concentration, with a linear dependence at lower monomer concentrations, followed by a saturation phase at higher monomer concentrations. The simulation in (**b**) (dashed line) uses a slower microscopic elongation rate ($k_e$) (*Table 4*) than that used in panel (**d**) and therefore saturation is not achieved by 410 µM protein in (**b**), but is in (**d**). Five replicates are shown for each protein concentration. Error bars show the standard deviation between all replicates.

DOI: https://doi.org/10.7554/eLife.46574.002

these regions form the intermolecular interfaces in the higher molecular weight species (*Figure 2d, e*). Consistent with these observations, measurement of the rotational correlation time ($\tau_c$) and diffusion coefficient of the sample, which reflect the average size and shape of the molecules formed, showed a linear dependence on $\Delta$N6 concentration, consistent with protein oligomerization in which

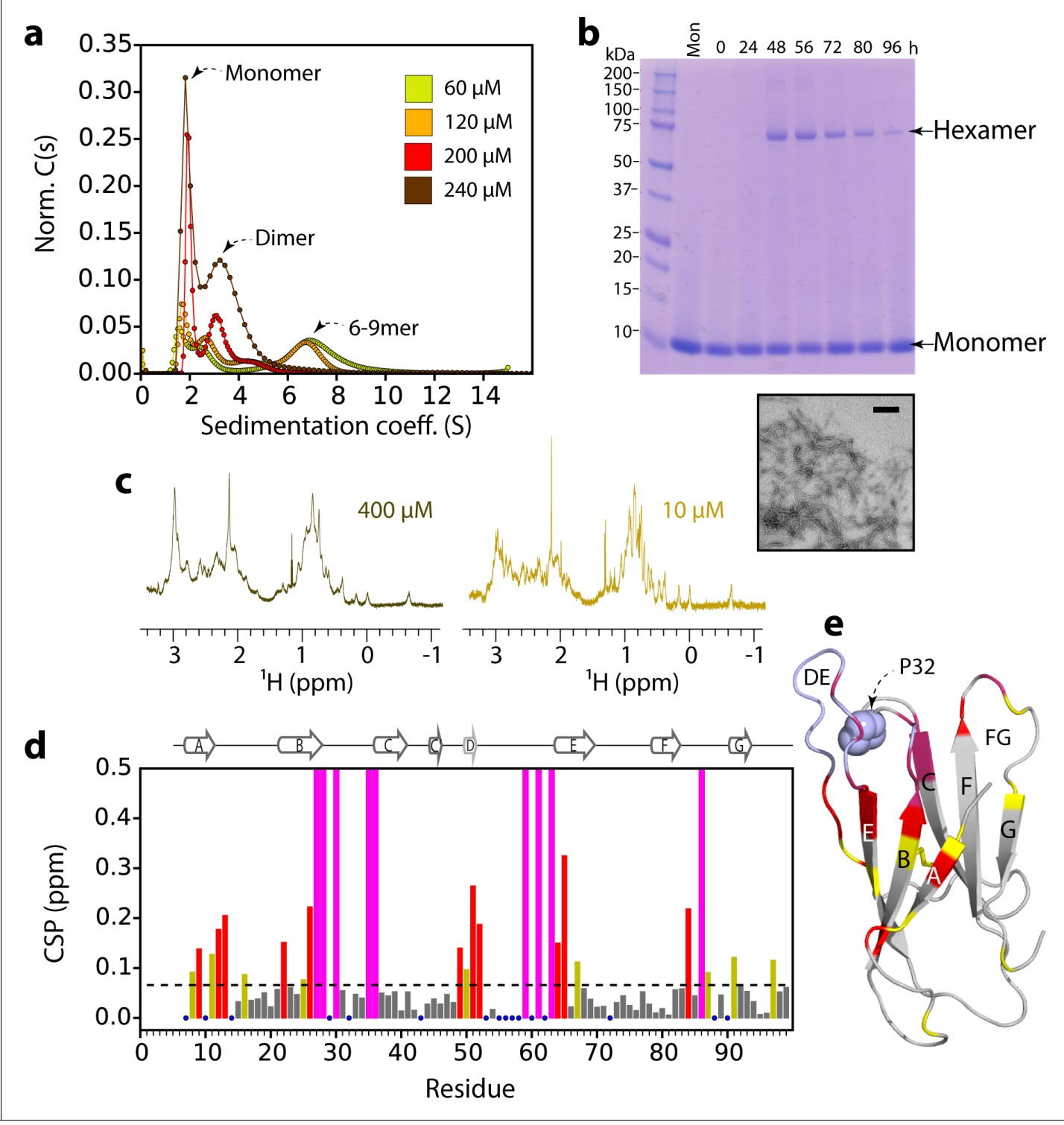

**Figure 2.** ΔN6 oligomer formation. (**a**) Sedimentation velocity AUC of ΔN6 at different concentrations, as indicated by the key. Note that the higher order species decrease in intensity at high protein concentrations (>200 μM) consistent with the formation of large aggregates that sediment rapidly before detection (see also *Figure 2—figure supplement 1d*). (**b**) SDS–PAGE of cross-linked ΔN6 (80 μM) at different time-points during de novo fibril assembly in the absence of fibril seeds (see Materials and methods). Note that dimers are not observed, presumably as they are not resilient to the vigorous agitation conditions used to accelerate fibril formation in these unseeded reactions, or are not efficiently cross-linked by EDC under the conditions used (see Materials and methods). A negative stain electron micrograph of ΔN6 after 100 hr of incubation is shown below. Scale bar – 500 nm. (**c**) The methyl region of the $^1$H NMR spectrum of ΔN6 at 400 μM (left) or 10 μM ΔN6 (right). (**d**) Per residue combined $^1$H-$^{15}$N chemical shift

*Figure 2 continued on next page*

*Figure 2 continued*

differences between the $^1$H-$^{15}$N HSQC spectrum of ΔN6 at 10 μM and 400 μM. Blue dots represent residues for which assignments are missing in both spectra. The dashed line represents one standard deviation (σ) of chemical shifts across the entire dataset. Residues that show chemical shift differences > 1σ are shown in yellow,>2σ are colored red, and residues for which the chemical shift difference is not significant (<1σ) are colored gray. Residues that are broadened beyond detection in the spectrum obtained at 400 μM are colored in magenta (see also *Figure 2—figure supplement 2a*). Residues are numbered according to the sequence of the WT protein. Arg 97 is hydrogen bonded to residues in the N-terminus and presumably is indirectly affected by the interaction. (**e**) The structure of ΔN6 (2XKU; *Eichner et al., 2011*) colored in the same scheme as (**d**). Pro32 is shown in blue space-fill. The buffer used in all experiments was 10 mM sodium phosphate pH 6.2 containing 83.3 mM NaCl (to maintain a constant ionic strength of 100 mM for all experiments), 25°C.

DOI: https://doi.org/10.7554/eLife.46574.003

The following figure supplements are available for figure 2:

**Figure supplement 1.** Analysis of ΔN6 oligomerization.
DOI: https://doi.org/10.7554/eLife.46574.004

**Figure supplement 2.** Estimation of dimer and hexamer $K_d$ values.
DOI: https://doi.org/10.7554/eLife.46574.005

the resulting species are in dynamic exchange (*Figure 2—figure supplement 1e,f*). Together these results show that ΔN6 assembles into dimers and hexamers that are assembly competent, in dynamic exchange, and assemble via interfaces which are located in the apical region of the protein that surrounds Pro32 (*Figure 2e*).

To estimate the dissociation constants for dimer and hexamer formation, the chemical shifts and residual dipolar couplings (RDCs) of individual resonances were measured as a function of ΔN6 concentration from 10 to 410 μM (*Figure 2—figure supplement 2a–e*). Significant chemical shift differences were observed when the ΔN6 concentration was increased from 10 μM to 50 μM without significant line broadening (*Figure 2—figure supplement 2c*, panels i-iii). Increasing the protein concentration to 100 μM caused a decrease in the chemical shift differences (*Figure 2—figure supplement 2c*, panel iv), which then increase again in magnitude at 200 μM and 410 μM, accompanied by significant line broadening (*Figure 2—figure supplement 2a and c*, panels v,vi). This complex behavior is consistent with a monomer-dimer-hexamer equilibrium in which the monomers and dimers have different chemical shifts, while the chemical shifts of dimers and hexamers are similar (an assumption that is supported by our structural models, see below), and the exchange rate between monomers and hexamers is significantly faster than that between monomers and dimers. Therefore, the monomer-dimer equilibrium dominates the equilibrium (and the observed chemical shift) at low concentrations (50 μM). At higher concentrations the dimer is depleted relative to the hexamer and the chemical shift observed becomes a complex combination of the population of each species, the exchange rate between each species, and the difference in chemical shift of each residue in each assembly. Fitting the chemical shift data to a monomer – dimer – hexamer model yields a $K_d$ for dimer formation of ≤50 μM, while that of hexamer formation is ~10 ± 5 x $10^{-9}$ M$^2$ (*Benilova et al., 2012*) (see Materials and methods and *Figure 2—figure supplement 2b,d*), indicating that once dimers form they have a high affinity for one another. Importantly, the monomer – dimer – hexamer model with the estimated affinities adequately describes the observed increase in the $\tau_c$ and the observed diffusion coefficient versus protein concentration (*Figure 2—figure supplement 1e,f*), independently supporting the model derived. Increasing the $K_d$ for dimer formation to >100 μM results in unrealistically low values for the hexamerization $K_d$ (*Figure 2—figure supplement 2d*). Moreover, measurement of RDCs versus protein concentration results in a biphasic curve (*Figure 2—figure supplement 2e*), consistent with a multi-equilibrium process. Using these data, the RDCs of the dimer species can be calculated for a range of estimated $K_d$ values (see Materials and methods). Fitting the dimer RDCs to the structure of ΔN6 (2XKU; *Eichner et al., 2011*), shows significantly poorer fits to the predicted RDC values assuming a dimer $K_d$ higher than 50 μM (*Figure 2—figure supplement 2f*). To explain the chemical shift and RDC data, therefore, the dimer $K_d$ must be ≤50 μM.

## Specific interfaces determine aggregation

To map the interfaces involved in ΔN6 oligomer formation in more detail, intermolecular paramagnetic relaxation enhancement (PRE) experiments were performed. The PRE depends on the distance

between a paramagnet and adjacent nuclei and can provide distance information about (transient) binding interfaces for nuclei that are within ~20 Å of the paramagnetic center (*Clore and Iwahara, 2009*), quantified by the effect of the spin label on the relaxation rates of each amide proton (the $H_N$-$\Gamma_2$ PRE rate). $^{14}$N-$\Delta$N6 was spin-labeled with (1-oxyl-2,2,5,5-tetramethyl-D3-pyrroline-3-methyl) methanethiosulfonate (MTSL) by creating Cys variants at positions 20, 33, 54 or 61. Each protein (60 µM) was then mixed with $^{15}$N-$\Delta$N6 (60 µM) at pH 6.2. At this total protein concentration, the PREs observed are dominated by the monomer-dimer equilibrium (35% of molecules are monomer, 51% of $\Delta$N6 molecules are in dimers and 14% of $\Delta$N6 molecules are in hexamers, determined from the $K_d$ values measured above). These experiments (*Figure 3*) showed increased $H_N$-$\Gamma_2$ rates for residues in the A strand and the BC, DE and FG loops when the spin label is attached to residues 33, 54, or 61, suggestive of a head-to-head interaction involving the apical regions of the protein (*Figure 3a–c and e*). In accord with this conclusion, when MTSL is attached at position 20 at the distal side of the protein (*Figure 3e*), the $H_N$-$\Gamma_2$ rates are vastly decreased (*Figure 3d*).

To determine whether the head-to-head dimers are critical for aggregation, the AUC, PRE and fibril growth experiments were also performed at pH 8.2 where $\Delta$N6 does not assemble into amyloid fibrils even after extended incubation times (*Figure 3—figure supplement 1a*). The sedimentation velocity AUC experiments revealed that monomers and tetramers are formed at pH 8.2, but not hexamers, with the equilibrium in favor of the monomer (*Figure 3—figure supplement 1b*). Consistent with this, the $\tau_c$ of 600 µM $\Delta$N6 at pH 8.2 is ~12 ns, in marked contrast with the $\tau_c$ of ~50 ns predicted for 600 µM $\Delta$N6 at pH 6.2 (*Figure 2—figure supplement 1e*). Finally, intermolecular PRE experiments at pH 8.2 showed small $\Gamma_2$ rates irrespective of the site of MTSL labeling (*Figure 3—figure supplement 1c–d*), suggesting that the monomers bind with different affinity and/or via different interfaces at this pH. To investigate these hypotheses, CPMG relaxation dispersion NMR experiments were performed. These experiments are able to detect excited states populated to as little as 1% of the total protein in solution (*Hansen et al., 2008*). Concentration-dependent CPMG profiles of residues in the B strand, D strand, DE loop, E strand and EF loop were observed at pH 8.2 (*Figure 3—figure supplement 2a–d*), indicating that the binding interface for tetramer formation differs substantially from the loop-loop interactions in the apical region of the protein that dominate assembly at pH 6.2, despite the fact that $\Delta$N6 retains an immunoglobulin-like fold at both pH values (*Figure 3—figure supplement 2e,f*). As a consequence of the altered interface that forms at pH 8.2, hexamers and fibrils do not form. Together these results indicate that the head-to-head dimers formed at pH 6.2 are uniquely able to assemble into the hexamers that are crucial for fibril assembly.

## Different dimer structures determine amyloid inhibition and propagation

To generate dimer structures consistent with the experimental data obtained, simulated annealing molecular dynamics calculations were performed. The calculations converged to two dimer structures (*Figure 4a*, *Figure 4—figure supplement 1* and *Table 1*). In the lowest energy model (model A), the $\Delta$N6 monomers are arranged in an extended conformation with the N-terminal residues M6 and I7 (WT numbering), along with the BC, DE and FG loops forming the interface (*Figure 4a*). The inhibitory dimer of $\Delta$N6:murine $\beta_2$m (m$\beta_2$m) was previously determined using a similar approach (*Karamanos et al., 2014*). This dimer also has a head-to-head subunit arrangement but is characterized by a more acute angle between $\Delta$N6 subunits in which the monomers interact predominantly through the BC and DE loops (*Karamanos et al., 2014*) (*Figure 4b*, *Video 1*). Thus, distinct protein dimers formed from closely related sequences (m$\beta_2$m and h$\beta_2$m are 70% identical and 90% similar in sequence) give rise to fundamentally different outcomes of assembly.

## Structural models of on-pathway hexamers

Although the majority of the intermolecular PREs can be satisfied by the dimer A structure, the fits are not perfect (*Figure 3a–d*), presumably since ~14% of $\Delta$N6 molecules form hexamers at the concentration of $\Delta$N6 employed (120 µM). The PRE experiments were thus repeated at higher concentrations (320–400 µM) of $\Delta$N6, wherein > 40% of $\Delta$N6 molecules are predicted to be in hexamers. These experiments revealed a pattern of $H_N$-$\Gamma_2$ rates similar to those obtained at 120 µM $\Delta$N6 (*Figure 5—figure supplement 1a–d*), with the highest $H_N$-$\Gamma_2$ rates involving the N-terminus, BC, DE and FG loops, suggesting that similar interfaces are formed in the dimer and hexamer. CPMG

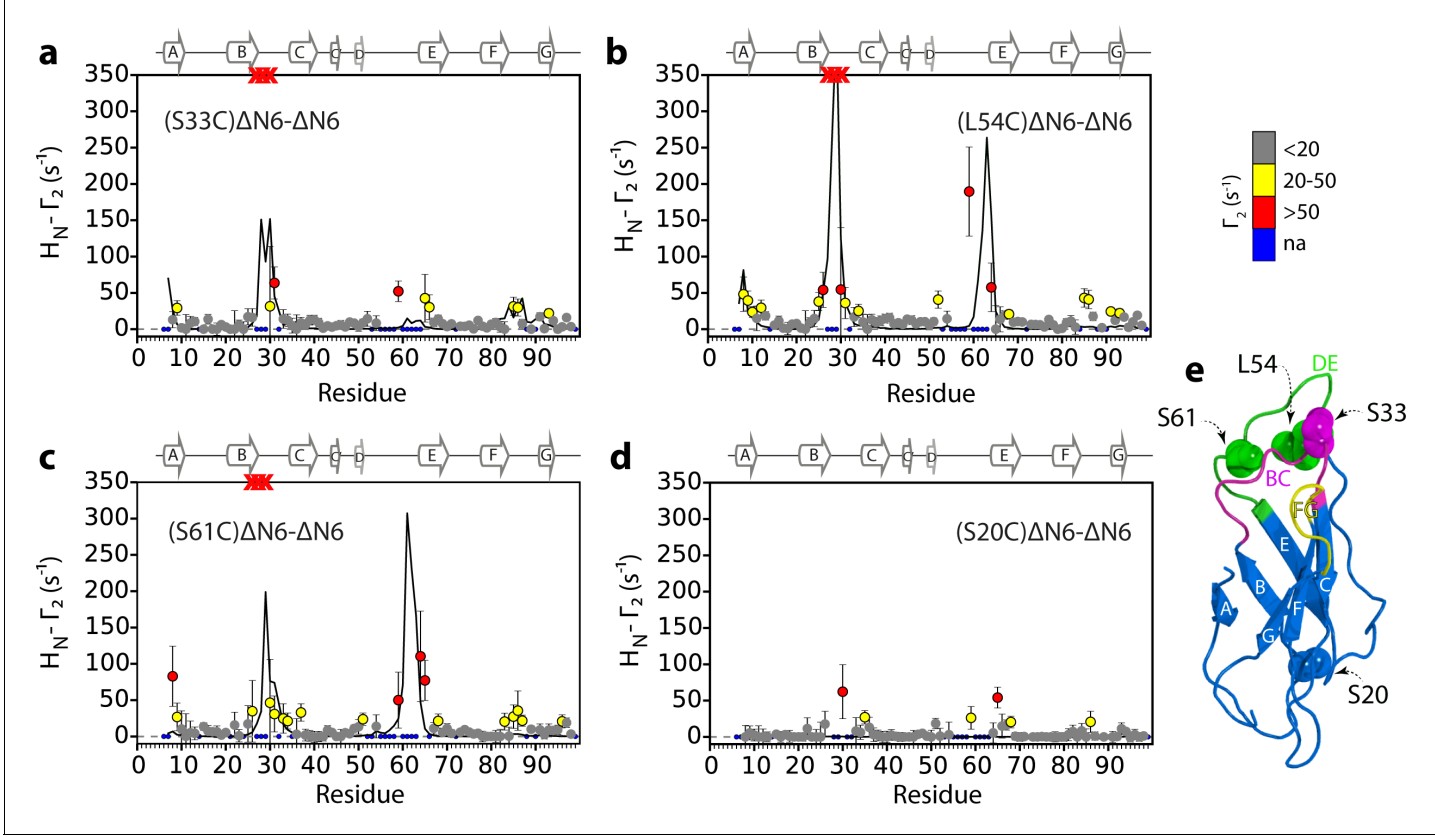

**Figure 3.** Identification of interacting surfaces in ΔN6 dimers. Intermolecular PRE data for the self-association of ΔN6. [15]N-ΔN6 (60 μM) was mixed with an equal concentration of (a) [14]N-(S33C)ΔN6-MTSL; (b) [14]N-(L54C)ΔN6-MTSL; (c) [14]N-(S61C)ΔN6-MTSL; or (d) [14]N-(S20C)ΔN6-MTSL in 10 mM sodium phosphate buffer, pH 6.2 containing 83.3 mM NaCl (a total ionic strength of 100 mM). The resulting $\Gamma_2$ rates are color-coded according to the amplitude of the PRE effect (see scale bar: gray-insignificant ($<20$ s$^{-1}$), yellow->20 s$^{-1}$, red->50 s$^{-1}$, pH 6.2, 25˚C). Blue dots in the plots are residues for which resonances are not assigned (na) at pH 6.2. Red crosses indicate high $H_N$-$\Gamma_2$ rates for which an accurate value could not be determined. Control experiments showed that the small PREs arising from[14]N-(S20C)ΔN6-MTSL arise from non-specific interactions with MTSL itself. Solid black lines depict fits to the PRE data for the dimer structure shown in *Figure 4a*. Note the poor fits for some residues which are sensitive to hexamer formation (14% of ΔN6 molecules) under the conditions used. Residues are numbered according to the WT sequence and the position of β-strands (2XKU; *Eichner et al., 2011*) is marked above each plot. (e) The structure of ΔN6 (2XKU; *Eichner et al., 2011*) with the BC loop shown in magenta, the DE loop in green and the FG loop in yellow. The MTSL attachment sites are highlighted as spheres.

DOI: https://doi.org/10.7554/eLife.46574.006

The following figure supplements are available for figure 3:

**Figure supplement 1.** Lack of a hexamer population precludes aggregation of ΔN6 at pH 8.2.

DOI: https://doi.org/10.7554/eLife.46574.007

**Figure supplement 2.** Mapping the interface of ΔN6 self-association at pH 8.2 using CPMG experiments.

DOI: https://doi.org/10.7554/eLife.46574.008

experiments at 180 μM ΔN6 (26% of ΔN6 molecules are monomer, 48% are dimer, and 26% are hexamer) and 480 μM ΔN6 (13% of ΔN6 molecules are monomer, 32% are dimer and 55% are hexamer) showed that residues in the apical regions of ΔN6, surrounding Pro 32, are also in concentration-dependent exchange at both ΔN6 concentrations at pH 6.2, in support of this conclusion (*Figure 5—figure supplement 2*).

The ordered nature of assembly (monomer, dimer, hexamer) and the identification of the interfaces involved, allowed us to generate models for the hexameric species (*Figure 5—figure supplement 3*). The measured PREs were converted into distances and simulated annealing molecular dynamics calculations were performed to create hexamer structures consistent with the experimental PRE and chemical shift data using the lowest energy dimer model (dimer A shown in *Figures 4a* and *5 - Figure 5—figure supplement 3a*), as well as the less favorable dimer model B (*Figure 4—figure*

*supplement 1e* and *Figure 5—figure supplement 3b*), as starting points. Note that the structure calculation strategy employed does not require knowledge of the dimer and hexamer populations (see Materials and methods). Starting from dimer A (*Figure 4a*) the structure calculation resulted in a hexamer in which the three dimers trimerize to form a compact daisy-like structure (*Figure 5a–c*). The PREs back-calculated from this model are consistent with the experimental data (*Figure 5—figure supplement 4*). Importantly, hexamer structures generated from dimer B (*Figure 4—figure supplement 1e*) resulted in poorer fits to the PRE profiles (Materials and methods and *Table 2*).

In the hexamer models generated from dimer A the dimeric subunits are arranged in a helical manner twisted by ~120°, creating a hexamer that is ~60 Å in diameter and 75 Å in length. This hexamer model is consistent with the collision cross-section (CCS) of ΔN6 hexamers measured using the lowest charge state (15+) (the most native-like species; *Vahidi et al., 2013*; see Materials and methods) detected using Electrospray Ionization Ion Mobility Spectrometry – Mass Spectrometry (ESI-IMS-MS), but the measured CCS is inconsistent with hexamers derived using dimer B (*Figure 5—figure supplement 5a*). The monomer-monomer and dimer-dimer interfaces in the best fit hexamer structure (*Figure 5a–e*) involve similar, but not identical, regions, with the inter-dimer interface extending further into the β-sheet containing the A, B, E and D β-strands, while the intra-dimer interface is formed mostly through the BC and DE loops (*Figure 5d–e* and *Video 2*). The formation of dimers generates a hydrophobic surface which becomes buried in the hexamer (*Figure 5e*, *Figure 5—figure supplement 5b* and *Table 3*). Consistent with this, the cross-linked hexamers show a small (1.3-fold) increase in fluorescence in the presence of the hydrophobic dye 8-anilino-1-naphthalenesulfonic acid (ANS), that is much smaller than the ~100 fold increase in ANS fluorescence observed for a typical 'molten globule' state (*Semisotnov et al., 1991*), but similar in magnitude to ANS bound to the highly structured on-pathway folding intermediate of Im7 (monitored using the trapped equilibrium mimic of this species, Im7 L53AI54A (*Spence et al., 2004*) (*Figure 5—figure supplement 5d*). The ΔN6 dimers show a similar increase in ANS fluorescence as the hexamers despite having a larger exposed hydrophobic surface area, possibly because ANS binds more weakly or has a lower quantum yield when dimer-bound (*Figure 5—figure supplement 5c–d*). The interface formed in the inhibitory ΔN6-mβ₂m dimer overlaps with the surface required for hexamerization, but not for ΔN6-ΔN6 dimerization (*Figure 5f*), rationalizing why mβ₂m is able to inhibit amyloid formation (note that the $K_d$ of the mβ₂m:ΔN6 complex is 70 μM (*Karamanos et al., 2014*), similar to that (~50 μM) estimated here for ΔN6 homo-dimerization). The dimers and hexamers were incubated with SH-SY5Y cells, a cell line that is commonly used in studies of amyloid toxicity (*Laganowsky et al., 2012*; *Fusco et al., 2017*; *Campioni et al., 2010*; *Jakhria et al., 2014*; *Leri et al., 2016*; *Giorgetti et al., 2008*), and which has been shown previously to take up monomeric and fibrillar β₂m (*Jakhria et al., 2014*). Interestingly, there was no evidence for cytotoxicity in assays for 3-(4,5-dimethylthiazol-2-yl)−2,5-diphenyltetrazolium bromide (MTT) reduction, lactate dehydrogenase release, reactive oxygen species formation and cellular ATP level (see Materials and methods) (*Figure 5—figure supplement 6*). However, rapid dissociation of the uncross-linked oligomers, prevention of conversion to a cytotoxic form by cross-linking, or cytotoxicity requiring different cell types or prolonged exposure (>24 hr) to the oligomers cannot be ruled out.

## Hexamer dynamics may prime further assembly into amyloid

The hexamer shown in *Figure 5* retains a native-like immunoglobulin fold in which the β-strands are anti-parallel. Hence, a major conformational rearrangement has still to occur for ΔN6 to form amyloid fibrils in which the β-strands stack in a parallel in-register structure (*Debelouchina et al., 2010*) (R. Silvers, Y. Su, R.G. Griffin, and S.E. Radford, unpublished). Hints of how this conformational change may be initiated were obtained by quantitative analysis of the CPMG data shown in *Figure 6*, *Figure 5—figure supplement 2* and *Figure 6—figure supplement 1*. Globally fitting these data for residues which lie in the dimer and/or hexamer interfaces (residues 26, 34, 35, 37, 51, 59, 65, 66, 67, 83, 84, 85, *Figure 5d–e*) to a two-state fast exchange model yields an exchange rate, $k_{ex}^{bind}$, of 1790 ± 290 s$^{-1}$ (*Figure 6a,b*). Distinct CPMG profiles were observed, however, for residues 87, 89, 91 and 92 which lie in the G strand of monomeric ΔN6 and which are not involved in the dimer-dimer interfaces (they show no significant concentration-dependent chemical shifts, nor PREs are observed for these residues at low or high protein concentration (*Figure 6—figure supplement 1a*). The CPMG data for these residues presumably report on conformational changes that result from

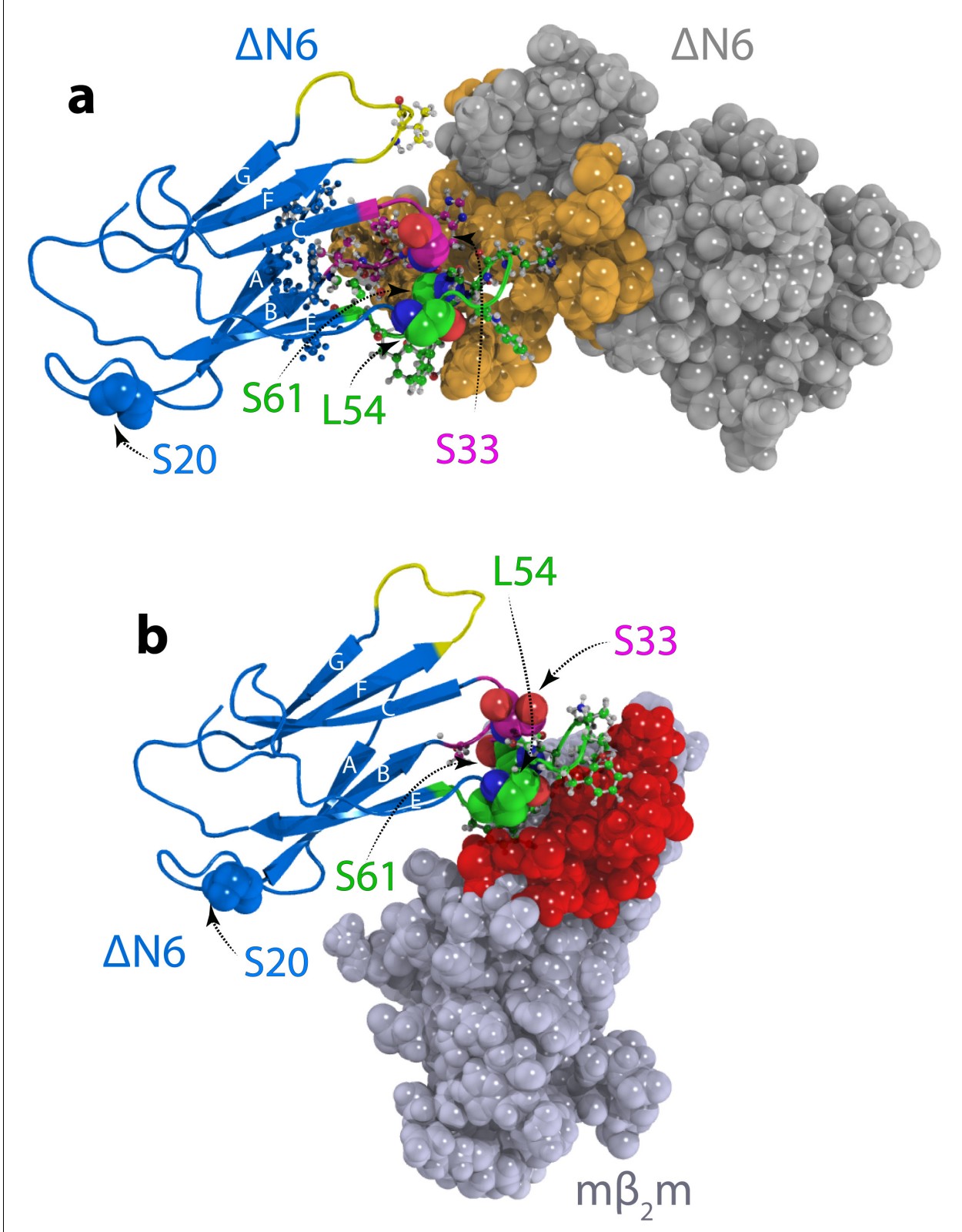

**Figure 4.** Structural models of ΔN6 dimers. Structural models of (**a**) the lowest energy ΔN6 homodimer (dimer A) and (**b**) the ΔN6-mβ$_2$m heterodimer that inhibits ΔN6 fibril assembly (*Karamanos et al., 2014*). Interface residues (identified as those residues that have any pair of atoms closer than 5 Å) are shown in a ball and stick representation on one subunit and are colored in space fill in gold in (**a**) or red in (**b**) on the surface of the second subunit. ΔN6 is shown in the same pose (blue) in (**a**) and (**b**). The BC, DE and FG loops are shown in magenta, green and yellow, respectively, and the position
*Figure 4 continued on next page*

*Figure 4 continued*

of attachment of MTSL for the PRE experiments (residues 20, 33, 54 and 61) is highlighted in spheres. PDB files are publicly available from the University of Leeds depository (https://doi.org/10.5518/329). See also ***Video 1***.

DOI: https://doi.org/10.7554/eLife.46574.009

The following figure supplement is available for figure 4:

**Figure supplement 1.** Alternative ΔN6 dimer structures.

DOI: https://doi.org/10.7554/eLife.46574.010

hexamerization rather than the direct binding event itself. The CPMG data indicate that these residues exchange with a lowly populated (2%) excited state with an interconversion rate, $k_{ex}^G$, of 205 ± 150 s$^{-1}$, 10-fold slower than $k_{ex}^{bind}$ (***Figure 6c–d*** and ***Figure 6—figure supplement 1a–e***). Therefore, a distinct process, possibly local unfolding of the C-terminal β-strand, occurs when the hexamer is formed that is driven by the free energy of hexamer formation ($\Delta G^{\circ}_{hexamer}$ ~4 kJ/mol). At ΔN6 concentrations of 480 μM $k_{ex}^G$ is increased to 1170 ± 196 s$^{-1}$ (***Figure 6—figure supplement 1f–i***), consistent with increased hexamer formation enhancing the observed rate of dynamics of the G strand. Hexamer formation thus potentially destabilizes the G-strand of ΔN6, causing local unfolding of this region of the polypeptide chain (although further experiments measuring the sign of the chemical shift change would substantiate this conclusion). This may then lead to more catastrophic structural reorganization of the hexamer into the parallel in-register structure of amyloid (note that the G-strand sequence forms a β-strand in the ΔN6 fibril core; ***Su et al., 2014***). Whether structural conversion occurs within the hexamer, at the fibril end, or requires further, more elaborate molecular steps such as active participation of the fibril surface, or disassembly into smaller structural units prior to fibril assembly, remains to be seen.

## A unified model of Δn6 polymerization

As a final test of the validity of the model of ΔN6 assembly proposed we assessed the ability of the structural, kinetic and thermodynamic parameters deduced above to describe the observed rates of fibril formation measured using ThT fluorescence, as well as the $\tau_c$ values versus ΔN6 concentration measured by NMR, and the fibril yield. Using the dimer and hexamer structural models shown in ***Figures 4*** and ***5*** and the estimated $K_d$ values for their formation, all of the derived experimental data could be recapitulated (***Figure 7***). Fitting the seeded fibril growth data to different kinetic models that assume (i) monomers to add to the fibril ends (***Figure 7—figure supplement 1a***); (ii) monomers are in exchange with a monomeric excited state that is responsible for growth (***Figure 7—figure supplement 1b***); or (iii) dimers are the elongation units (***Figure 7—figure supplement 1c***), fail to describe the seeding data (Materials and methods and ***Table 4***). By contrast, a model assuming addition of hexamers describes the ThT kinetic profiles well (***Figure 7a***), while a more complicated monomer-dimer-tetramer-hexamer model does not improve the fit significantly (***Figure 7—figure supplement 1d***). The populations of monomer, dimer and hexamer, together with the derived

**Table 1.** Agreement between experimental and back-calculated intermolecular PREs for the two dimer structures (dimer A and dimer B (see ***Figure 5—figure supplement 3***).

RMS values are shown comparing the measured versus the predicted values from the structure PREs measured from S33, L54 and S61. Data from position S20 were not used as they arise from non-specific interactions with MTSL.

| PRE term | RMS dimer A | RMS dimer B |
|---|---|---|
| S33C(ΔN6)-ΔN6 (s$^{-1}$) | 18.65 | 15.10 |
| L54C(ΔN6)-ΔN6 (s$^{-1}$) | 29.02 | 27.44 |
| S61C(ΔN6)-ΔN6 (s$^{-1}$) | 19.44 | 23.27 |
| *High PREs (Å) | 2.78 | 3.79 |

*High PREs refer to PREs in the BC loop (measured from S33, L54 and S61) that (due to their large value) could not be measured accurately and therefore are incorporated as loose distance restraints.

DOI: https://doi.org/10.7554/eLife.46574.011

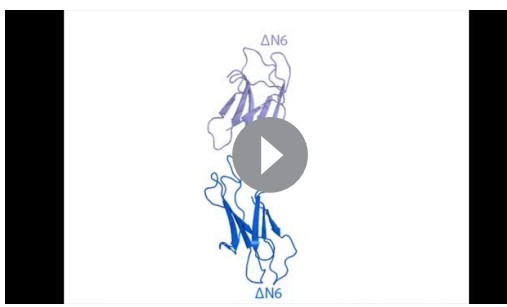

**Video 1.** Comparison of productive and inhibitory dimers. The ΔN6 subunit in each dimer (this study) is shown as a dark blue cartoon, while the second ΔN6 monomer in the productive dimer and the mβ₂m subunit in the inhibitory dimer (*Karamanos et al., 2014*) are shown as light blue and red, respectively. The BC, DE, FG loops are colored in magenta, green and yellow, respectively, while the intra-dimer interface residues are shown as sticks on both subunits.
DOI: https://doi.org/10.7554/eLife.46574.012

structural models, are also consistent with the observed dependence of $\tau_c$ on protein concentration (*Figure 7b*). Finally, the amount of hexamer formed (in the absence of seeds) is also predictive of the fibril yield (*Figure 7c,d*) consistent with the hexamer being required for fibril formation. This conclusion is also supported by the appearance of hexamers early during assembly in the absence of seeds and their disappearance as fibrils form (*Figure 2b*).

## Discussion

Understanding the molecular details of oligomer formation is vital if we are to understand why proteins aggregate into amyloid and why different species have different toxicities (*Iadanza et al., 2018a*; *Lu et al., 2013*). Here, we present a general strategy, summarized in *Figure 8—figure supplement 1*, which allows the identification of oligomeric intermediates in amyloid assembly and enables their structural characterization. By combining the powers of NMR to detect lowly populated species in dynamic exchange, with complementary techniques such as AUC and cross-linking, oligomeric intermediates can be identified and structurally characterized in atomic detail. Importantly, to link these intermediates to the mechanism of aggregation, the derived affinities, stoichiometries and structural models can then be used to globally model the time course of fibril assembly. The strategy presented is not only applicable to protein aggregation, but to any weakly self-associating protein system. Given that the balance between monomers, dimers, higher molecular weight oligomers and fibrils could depend critically on the experimental conditions, including the pH, temperature, protein concentration, amount of seed added, buffer composition and ionic strength, the same protein, or a closely related protein variant, could assemble via different mechanism(s) under different conditions. Indeed, aggregation of many amyloidogenic proteins, including hβ₂m (*Iadanza et al., 2018b*), is known to result in polymorphic fibrils (*Close et al., 2018*; *Fitzpatrick et al., 2017*; *Colvin et al., 2016*; *Zhang et al., 2019*) that could extend via different mechanisms. The approach described here can distinguish between such different assembly pathways and may be able to shed light on the role of individual oligomeric species in aggregation and the origins of amyloid polymorphism.

Using the workflow derived, we show that elongation of ΔN6 amyloid seeds proceeds via a specifically organized hexamer (*Figure 8*). This finding contrasts with the more common view of monomer addition to fibril ends that has been observed for Aβ40/42 (*Cohen et al., 2018*), α-synuclein (*Buell et al., 2014*), huntingtin exon 1 (*Vitalis et al., 2009*) and for unfolded hβ₂m at pH 2.0 (*Figure 1a*) (*Xue et al., 2008*), while oligomers are thought to play critical roles in the primary/secondary nucleation phases of the assembly of these proteins (*Cohen et al., 2018*). By contrast with these initially disordered proteins, the monomeric precursor of ΔN6 assembly is structured, a scenario that accounts for more than 20 of the 70 human proteins known to cause amyloid disease (*Sipe et al., 2016*). Other amyloid precursors that are initially structured show an inability to self-seed (e.g. transthyretin; *Hurshman et al., 2004*), or display a non-classical dependence of the elongation rate on protein concentration (e.g. light chains; *Blancas-Mejía et al., 2017*). Whether these and other structured protein precursors assemble by a mechanism akin to that of ΔN6 could be answered by applying the integrated kinetic and structural approach described here to further examples of this set of proteins.

Here, we show that ΔN6 dimers and hexamers with well-defined interfaces involving the apical regions of the protein are required for fibrils to form under the conditions employed (*Figure 8*). By contrast, formation of other interfaces, such as that observed here for ΔN6 at pH 8.2 and the previously reported mβ₂m:ΔN6 heterodimer (*Karamanos et al., 2014*) are not able to assemble into

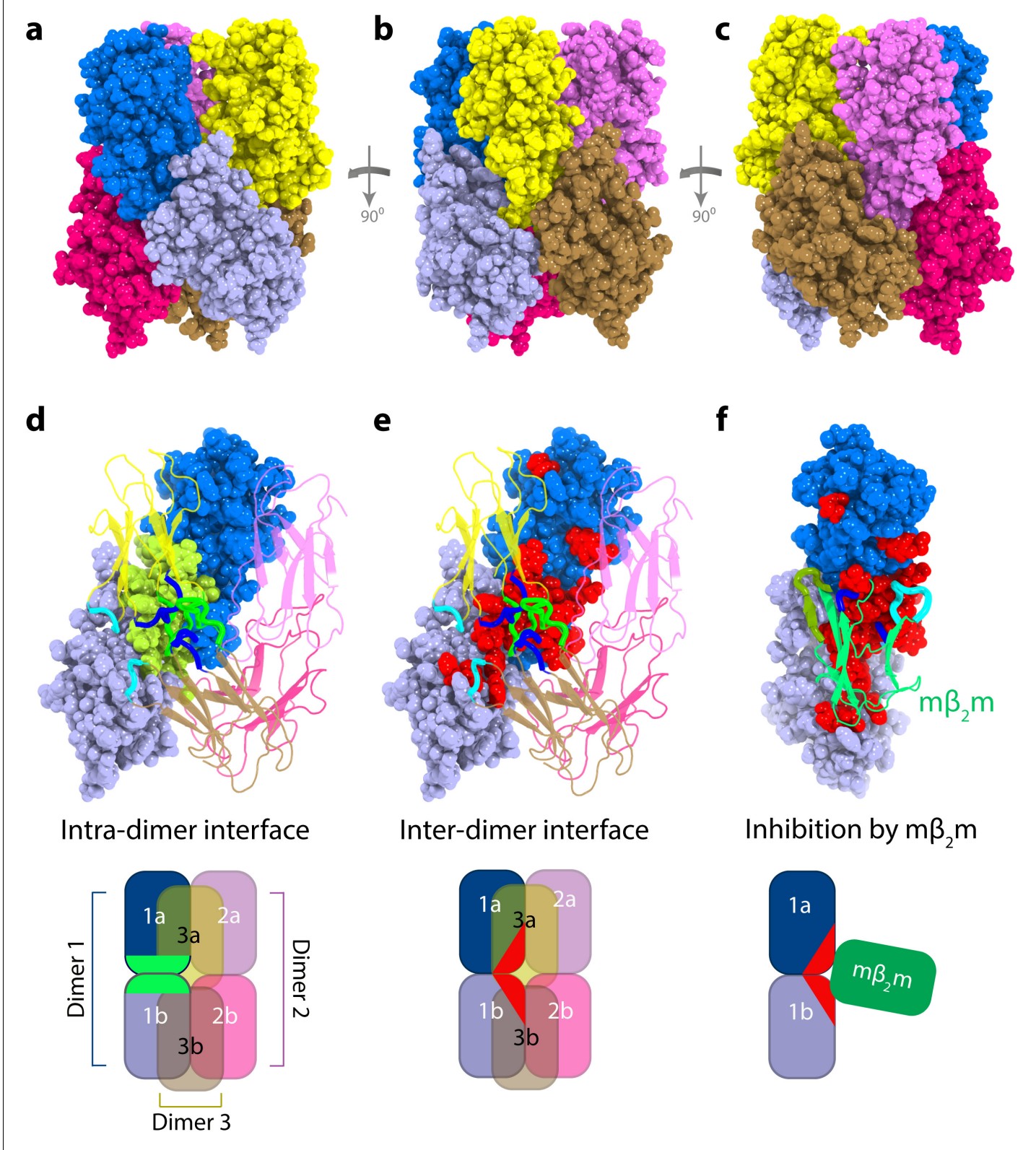

**Figure 5.** Structural model of ΔN6 hexamers. (**a–c**) Sphere representations of the hexamer model formed from dimer A rotated by 90° in each view. Subunits belonging to the same dimer are colored in different tones of the same color. (**d**) The monomer-monomer (intra-dimer) interface is highlighted in green on the surface of the dimer formed from subunits 1a and 1b (within dimer A), with the other dimers shown as cartoons. (**e**) The inter-dimer interface is colored red on the surface of the dimer formed from subunits 1a and 1b, with the dimers shown as cartoons. (**f**) As in (**e**), but showing the

*Figure 5 continued on next page*

*Figure 5 continued*
dimer formed from subunits 1a and 1b, superposed with the mβ₂m subunit in the inhibitory ΔN6-mβ₂m dimer (**Karamanos et al., 2014**) (green cartoon). The ΔN6-ΔN6 and ΔN6-mβ₂m dimers were aligned on the ΔN6 subunit 1b. Schematics of the assemblies are shown at the bottom colored as in (**d–f**). Note that the BC, DE and FG loops are highlighted as thicker chains in blue, green and cyan, respectively, in d-f. PDB files are publicly available from the University of Leeds depository (https://doi.org/10.5518/329). See also **Video 2**.
DOI: https://doi.org/10.7554/eLife.46574.013

The following figure supplements are available for figure 5:

**Figure supplement 1.** Intermolecular PREs at high ΔN6 concentration.
DOI: https://doi.org/10.7554/eLife.46574.014

**Figure supplement 2.** Additional interfaces do not form in the ΔN6 hexamer.
DOI: https://doi.org/10.7554/eLife.46574.015

**Figure supplement 3.** Initial docking of dimer structures to create hexamer models.
DOI: https://doi.org/10.7554/eLife.46574.016

**Figure supplement 4.** Intermolecular PREs back-calculated from the hexamer structural model generated from dimer A.
DOI: https://doi.org/10.7554/eLife.46574.017

**Figure supplement 5.** Conformational and biochemical properties of ΔN6 hexamers.
DOI: https://doi.org/10.7554/eLife.46574.018

**Figure supplement 6.** ΔN6 oligomers are not cytotoxic to SH-SY5Y cells.
DOI: https://doi.org/10.7554/eLife.46574.019

amyloid fibrils (**Figure 8**). The arrangement of subunits in the ΔN6 dimer and hexamer observed here is different to that in a previously reported structure of a domain swapped ΔN6 dimer (**Figure 8—figure supplement 2c**). However, the G strand that is responsible for the domain swap is dynamic in the hexamer structure presented here, consistent with this edge β-strand being able to dissociate from the β-sandwich to form both structures. A variant (H13F) of hβ₂m has also been reported to form hexamers in the presence of $Cu^{2+}$ ions (**Calabrese et al., 2008**) (**Figure 8—figure supplement 2a,b**). In the crystal structure of this species, the dimers and hexamers interact in a side-to-side or head-to-head manner to create a ring-like assembly, in marked contrast with the daisy-like organization of monomers in the ΔN6 hexamers shown in **Figure 5**. Real-time NMR studies of the folding of hβ₂m have also revealed protein concentration-dependent exchange-broadening in the apical loops of its transient folding intermediate $I_T$ (**Rennella et al., 2013**), an observation that has been attributed to head-head oligomers, in agreement with the data presented here for ΔN6 which structurally mimics $I_T$ (**Eichner et al., 2011**). The interfaces observed in the ΔN6 dimer and hexamer also differ from the canonical inter-sheet stacking between immunoglobulin domains in antibodies, suggesting that the structural features described here are specific to the dimers and hexamers involved in amyloid assembly. Taken together, the results show that β₂m can form different protein-protein interactions, only a specific set of which results in species capable of assembly into amyloid.

Although many studies have attributed the toxicity of amyloid to oligomeric species (**Chiti and Dobson, 2017**), we show here that the dimers and hexamers of ΔN6 are not cytotoxic, at least under the conditions employed, possibly because they are structured and bury substantial hydrophobic

**Table 2.** Agreement between experimental and back-calculated intermolecular distances for different hexamer structures.
RMS values are shown comparing the measured versus the predicted distances from each structural model for distances measured from S33, L54 and S61. Data from position S20 were not used as they arise from non-specific interactions with MTSL. See also **Figure 5—figure supplement 3**.

| PRE term | Hexamer 1 RMS (Å) | Hexamer 2(i) RMS (Å) | Hexamer 2(ii) RMS (Å) | Hexamer 2(iii) RMS (Å) |
|---|---|---|---|---|
| S33C(ΔN6)-ΔN6 | 2.34 | 2.68 | 2.58 | 2.53 |
| L54C(ΔN6)-ΔN6 | 1.25 | 2.33 | 2.26 | 1.87 |
| S61C(ΔN6)-ΔN6 | 2.22 | 2.7 | 2.68 | 3.11 |

DOI: https://doi.org/10.7554/eLife.46574.020

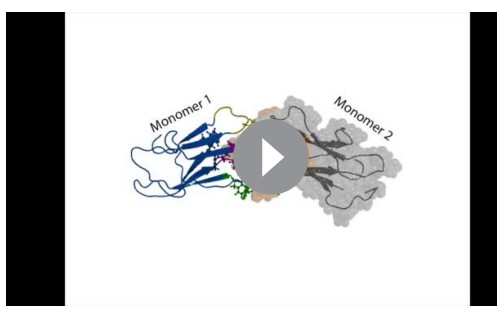

**Video 2.** ΔN6 assembles into dimers and hexamers. The two ΔN6 subunits in the dimer (dimer A) are shown as blue cartoon and gray cartoon/transparent space-filling representations, respectively. The BC, DE and FG loops are colored magenta, green and yellow, respectively. The intra-dimer interface residues are shown as sticks on one subunit and as orange transparent spheres on the second subunit. The hexamer assembly is then shown as a space-filling model, with dimer one shown in dark blue/light blue, dimer two in dark yellow/light yellow and dimer three in magenta/pink. In the last part of the video only dimer one is shown as spheres while dimers 2 and 3 are shown as transparent cartoons. The intra-dimer interface is shown in green and the inter-dimer interface is shown in red.

DOI: https://doi.org/10.7554/eLife.46574.022

surface area. Interestingly, the oligomerization of ΔN6 has been linked to increased toxicity in *Caenorhabditis elegans* models (*Diomede et al., 2012*). Since amyloid formation can proceed via multiple pathways, it is possible that the cytotoxic species of ΔN6 formed in the *C. elegans* body wall muscle are different to those formed here in vitro. For several proteins, cytotoxicity has been ascribed to off-pathway oligomers that accumulate in the lag time of assembly, consistent with amyloid formation being protective for the cell (*Bieschke et al., 2011*). Interconversion between different forms of oligomers may also be required for cytotoxicity (*Fusco et al., 2017*; *Cremades et al., 2012*). Such a process could be compromised in the cross-linked species of ΔN6 used here.

In summary, by taking advantage of the power of NMR spectroscopy to visualize transient species, and combining these experiments with detailed analysis of the kinetic, thermodynamic and hydrodynamic properties of the aggregating ensemble of species, we have been able to determine an atomic structural model of two oligomeric species required for amyloid formation of ΔN6 at pH 6.2, and have generated a model that describes a potential mechanism of fibril elongation from these states. Our findings portray an assembly mechanism that is remarkably well-defined, involving the formation of specific protein-protein interfaces that are unique to the initiating stages of amyloid assembly. Substantial conformational changes have still to occur, however, for the hexameric intermediate to form the cross-β structure of amyloid. How this is achieved remains an open question, but could involve binding to the fibril ends and/or fibril surfaces. Most importantly, the results reveal a remarkable specificity to the early stages of ΔN6 amyloid assembly that involves the formation of well-defined oligomeric species via specific interfaces, the precise details of which determine the course of assembly. These findings suggest new avenues to combat disease by specific targeting of the early intermediates in the amyloid cascade which, at least for ΔN6, involve specific interactions between non-native, assembly-competent states.

**Table 3.** Analysis of dimer and hexamer interfaces.
The buried surface area is calculated as the sum for the two subunits for each complex. Interface residues were identified as those residues that lose at least 10% of accessible surface area upon oligomer formation.

|  | ΔN6 dimer A | ΔN6 hexamer |
| --- | --- | --- |
| Buried Surface Area (Å$^2$) | 1233 | 4201 |
| % Charged residues in the interface | 28 | 18 |
| % Hydrophobic residues in the interface | 44 | 54 |

DOI: https://doi.org/10.7554/eLife.46574.021

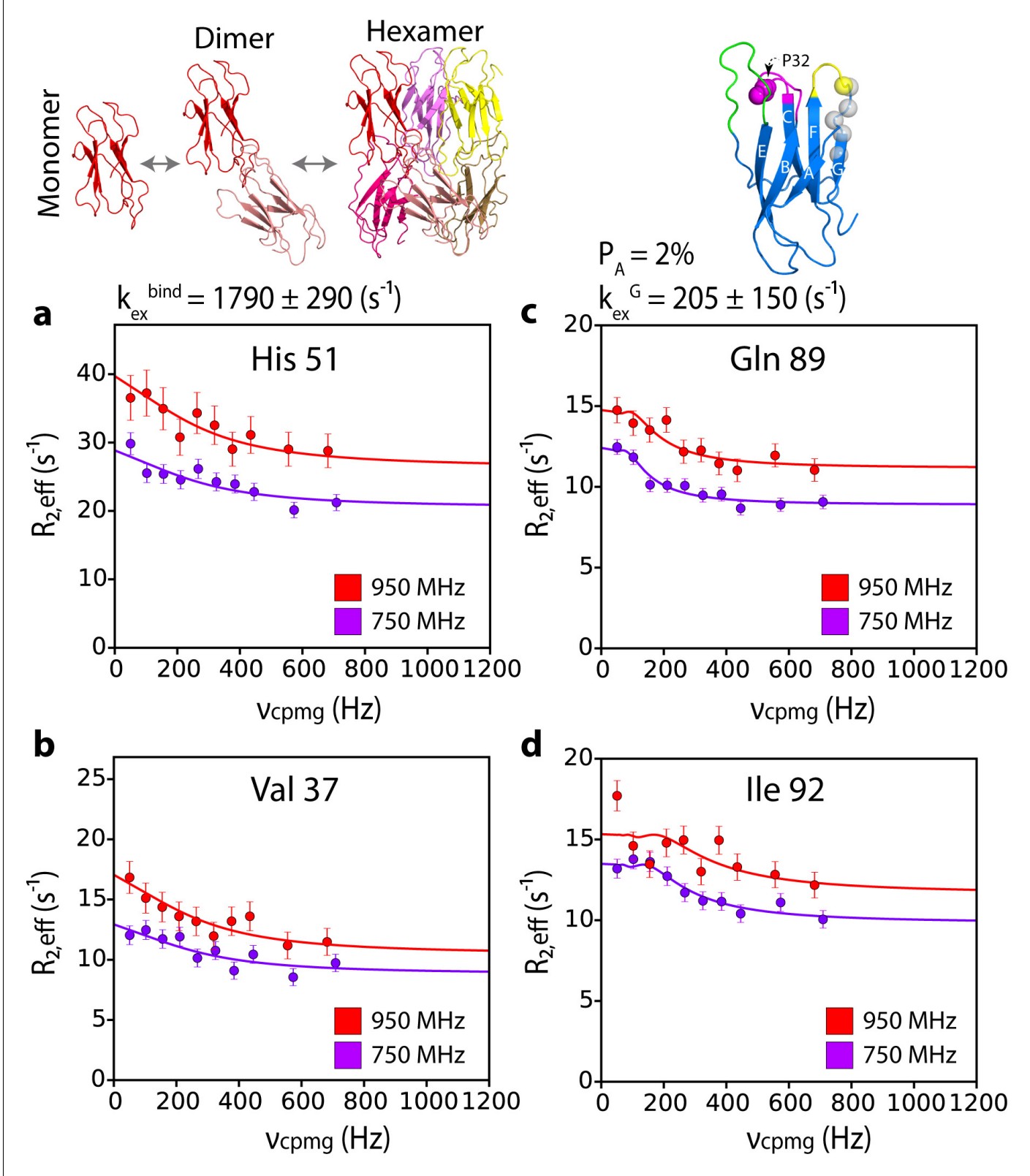

**Figure 6.** G-strand unfurling may occur upon hexamer formation. $^{15}$N CPMG relaxation dispersion data at 750 MHz (magenta) and 950 MHz (red) (180 μM ΔN6, pH 6.2 (26% ΔN6 molecules are monomers, 48% are in dimers, 26% are in hexamers) for residues (**a**) 51, (**b**) 37, (**c**) 89, and (**d**) 92. Residues 37 and 51 report on intermolecular interactions that describe dimer and/or hexamer formation (schematic, top left), while residues 89 and 92 do not lie in an interface and report instead in the dynamics of the G strand in the different assemblies formed. The position of all five residues used in the cluster

*Figure 6 continued on next page*

*Figure 6 continued*

analysis of G strand dynamics is shown in spheres on the structure of ΔN6 (blue cartoon, top right). Pro32 is shown as a magenta sphere. Solid lines represent global fits to the Bloch-McConnell equations (Materials and methods) for each cluster of residues. The extracted parameters of the global fit for the two processes ($k_{ex}^{bind}$ and $k_{ex}^{G}$) are indicated above the plots.

DOI: https://doi.org/10.7554/eLife.46574.023

The following figure supplement is available for figure 6:

**Figure supplement 1.** Hexamer formation increases the dynamics of the G strand.

DOI: https://doi.org/10.7554/eLife.46574.024

## Materials and methods

### Protein expression and purification

The pINK plasmid containing the ΔN6 gene was transformed into BL21 DE3 plysS *E. coli* cells. 2 L flasks containing 1 L of LB or HDMI (1 g/L $^{15}$N-NH$_4$Cl, 2 g/L $^{13}$C-glucose) media were inoculated with 10 mL of starter culture. Cells were incubated at 37°C, 200 rpm until they reached an OD$_{600}$ of ~0.6 and then the expression of ΔN6 was induced by the addition of 1 mM isopropyl β-D-1-thio-galactopyranoside (IPTG). Expression was allowed to continue overnight at 37°C and cells were harvested the next morning using a Heraeus continual action centrifuge at 15,000 rpm. The cell pellet containing ΔN6 as inclusion bodies was lysed by the addition of 50–100 mL buffer containing 100 µg/mL lysozyme, 50 µg/mL DNAse I, 50 µg/mL phenylmethanesulfonyl fluoride (PMSF), 10 mM Tris HCl pH 8.2. Further cell disruption was performed using a constant cell disrupter system (Constant Systems) at a pressure of 20.0 kpsi. Inclusion bodies were isolated using centrifugation (15,000 *g*) for 40 min, 4°C and the inclusion body pellet was washed with 10 mM Tris HCl pH 8.2 four times. ΔN6 was then solubilized in 10–20 mM Tris HCl pH 8.2 containing 8 M urea (MP Biomedicals) and refolded by dialysis (3000 MW cutoff membrane) against 2–5 L of the same buffer lacking urea. The refolded protein was centrifuged for 30 min at 15,000 *g* to pellet-insoluble material and the supernatant was loaded onto a Q-Sepharose (GE Healthcare) column equilibrated with 20 mM Tris HCl pH 8.2. Bound protein was eluted with a gradient of 0–400 mM NaCl in the same buffer over 800 mL and the protein was freeze-dried after dialysis in 18 MΩ H$_2$O or concentrated using 3000 MW cutoff centrifugal concentrators (Sartorius). The freeze-dried protein was re-suspended in 10 mM sodium phosphate buffer pH 7.0, filtered through 0.2 µm filters (Fisher Scientific) and gel filtered using a HiLoad Superdex-75 Prep column (Amersham Biosciences), calibrated with a standard gel filtration kit (GE Healthcare). The monomer peak was collected, concentrated, aliquoted and stored at −80°C or dialyzed into 18 MΩ H$_2$O and freeze-dried. Cys mutants of ΔN6 were created as described in reference (*Karamanos et al., 2014*) and purified as above, except that 2 mM dithiothreitol (DTT) was added before gel filtration.

### Aggregation assays

ΔN6 seeds were formed by incubation of 800 µM protein in 10 mM sodium phosphate buffer, pH 6.2 containing 83.3 mM NaCl (to give a total ionic strength of 100 mM), 0.02% (*w/v*) NaN$_3$ with 200 rpm shaking on a thriller shaker (Peqlab) at 37°C for 2 weeks. hβ$_2$m seeds were formed by incubation of 800 µM protein (expressed and purified as described in *Karamanos et al., 2014*) in 10 mM sodium phosphate buffer pH 2.0, containing 50 mM NaCl, 0.02% (*w/v*) NaN$_3$ with 200 rpm shaking at 37°C for 2 weeks. The resulting fibrils were sonicated for 15 s to create fibril seeds. For seeding reactions, samples containing 50–500 µM hβ$_2$m or ΔN6 in pH 2.0 or pH 6.2 buffers, respectively, containing 10 µM thioflavin T (ThT) were incubated quiescently at 37 °C in sealed 96 low binding well plates (Thermo Scientific). De novo ΔN6 fibrils were formed by incubating 60 µM ΔN6 in 10 mM sodium phosphate buffer, pH 6.2, containing 83.3 mM NaCl, 0.02% (*w/v*) NaN$_3$ with 600 rpm shaking in a 96-well plate at 37°C (lag time ~20 hr) or in an 1.5 mL Eppendorf tube (lag time ~100 hr). Control experiments monitoring seeded fibril growth of ΔN6 at pH 8.2 were performed in 10 mM sodium phosphate buffer, pH 8.2 containing 86.6 mM NaCl (total ionic strength 100 mM, identical to that used at pH 6.2) and 0.02% (*w/v*) NaN$_3$. Fluorescence was monitored at 480 ± 10 nm after excitation at 440 ± 10 nm using a FLUOROstar Optima micro-plate reader (BMG Labtech).

**Table 4.** Reaction schemes, rate equations and rate constants for the fibril elongation models tested. $X$ represents the species that add onto the fibril ends.

| Module | Variant | Reaction scheme | Rate equations | Rate constants |
|---|---|---|---|---|
| Pre-polymerization | No Pre-polymerization (Monomer addition) | $X = X_1$ | $\frac{d[X]}{dt} = \sum_{i=2}^{N} -k_e F_{i-1} X + k'_e F_i$ | $k_1 = k_e$ $k'_1 = k'_e,$ |
| | Monomer conformational exchange | $X_1 \underset{k'_1}{\overset{k_1}{\rightleftharpoons}} X'_1$ $X = X'_1$ | $\frac{d[X_1]}{dt} = -k_1 X_1 + k'_1 X'_1$ $\frac{d[X]}{dt} = \begin{cases} k_1 X_1 - k'_1 X'_1 \\ + \sum_{i=2}^{N} -k_e F_{i-1} X + k'_e F_i \end{cases}$ | $k_1, \ k_e$ $k'_1, \ k'_e,$ |
| | Dimer addition | $X_1 + X_1 \underset{k'_1}{\overset{k_1}{\rightleftharpoons}} X_2$ $X = X_2$ | $\frac{d[X_1]}{dt} = -2k_1 X_1 X_1 + 2k'_1 X_2$ $\frac{d[X]}{dt} = \begin{cases} k_1 X_1 X_1 - 2k'_1 X \\ + \sum_{i=2}^{N} -k_e F_{i-1} X + k'_e F_i \end{cases}$ | $k_1, \ k_e$ $k'_1, \ k'_e,$ |
| | Hexamer addition | $X_1 + X_1 \underset{k'_1}{\overset{k_1}{\rightleftharpoons}} X_2$ $X_2 + X_2 + X_2 \underset{k'_1}{\overset{k_1}{\rightleftharpoons}} X_6$ $X = X_6$ | $\frac{d[X_1]}{dt} = -2k_1 X_1 X_1 + 2k'_1 X_2$ $\frac{d[X_2]}{dt} = k_1 X_1 X_1 - - k'_1 X_2 - 3k_2 X_2 X_2 X_2 + 3k'_2 X_6$ $\frac{d[X]}{dt} = \begin{cases} k_2 X_2 X_2 X_2 - k'_2 X \\ + \sum_{i=2}^{N} -k_e F_{i-1} X + k'_e F_i \end{cases}$ | $k_1, \ k_2, \ k_e,$ $k'_1, \ k'_2, \ k'_e,$ |
| | Monomer-Dimer-Tetramer-Hexamer | $X_1 + X_1 \underset{k'_1}{\overset{k_1}{\rightleftharpoons}} X_2$ $X_2 + X_2 \underset{k'_2}{\overset{k_2}{\rightleftharpoons}} X_4$ $X_2 + X_4 \underset{k'_3}{\overset{k_3}{\rightleftharpoons}} X_6$ | $\frac{d[X_1]}{dt} = -2k_1 X_1 X_1 + 2k'_1 X_2$ $\frac{d[X_2]}{dt} = k_1 X_1 X_1 - - k'_1 X_2 - 2k_2 X_2 X_2 + 2k'_2 X_4 - k_3 X_4 X_2 + k'_3 X_6$ $\frac{d[X_4]}{dt} = k_2 X_2 X_2 - - k'_2 X_4 - k_3 X_4 X_2 + k'_3 X_6$ | $k_1, \ k_2, \ k_3 \ k_e$ $k'_1, \ k'_2, \ k'_3, k'_e,$ |
| | | $X = X_6$ | $\frac{d[X]}{dt} = \begin{cases} k_3 X_4 X_2 - k'_3 X \\ + \sum_{i=2}^{N} -k_e F_{i-1} X + k'_e F_i \end{cases}$ | |
| Polymerization | | $\begin{array}{cc} X & X \\ \downarrow & \downarrow \\ F \underset{k'_e}{\overset{k_e}{\rightleftharpoons}} F_1 \underset{k'_e}{\overset{k_e}{\rightleftharpoons}} F_2 \dots F_N \\ \downarrow & \downarrow \\ X & X \end{array}$ | $\frac{d[F]}{dt} = -k_e X F_1 + k'_e F_2$ $\frac{d[F_i]}{dt} = k_e X F_{i-1} - k'_e F_i - - k_e X F_i + k'_e F_{i+1} \quad 2 \le i < N$ $\frac{d[F_N]}{dt} = k_e X F_{i-1} - k'_e F_i$ | |

DOI: https://doi.org/10.7554/eLife.46574.027

## Analytical ultracentrifugation

For sedimentation velocity experiments, a sample of 450 μL of protein was dialyzed overnight with 10 mM sodium phosphate buffer, pH 6.2 containing 83.3 mM NaCl or 10 mM sodium phosphate buffer, pH 8.2 containing 86.6 mM NaCl (each buffer has a total ionic strength of 100 mM). The sample was inserted in double-sector Epon centerpieces equipped with sapphire windows and inserted in an An60 Ti four-cell rotor. Absorbance data at the appropriate wavelength were acquired at a rotor speed of 48,000 rpm at 25°C. Data were analyzed using the c(s) continuous distribution of the Lamm equations with the software SEDFIT (**Brown and Schuck, 2006**),

$$D(s) = \frac{\sqrt{2}}{18\rho} kT s^{-1/2} (\eta (f/f_0)_w)^{-3/2} ((1 - \bar{v}r)/\bar{v})^{1/2},$$

where $D(s)$ is the diffusion coefficient, $k$ Boltzmann's constant, $T$ the temperature in K, $s$ the sedimentation coefficient, $f$ the frictional coefficient, $f_0$ the frictional coefficient of a compact smooth sphere, $\eta$ the solvent viscosity, $\rho$ the solvent density and the partial specific volume.

At concentrations over 200 μM 20% of the material sedimented during the initial 3000 rpm run, consistent with the hexamers forming high-molecular-weight species that sediment before the c(S) data are acquired.

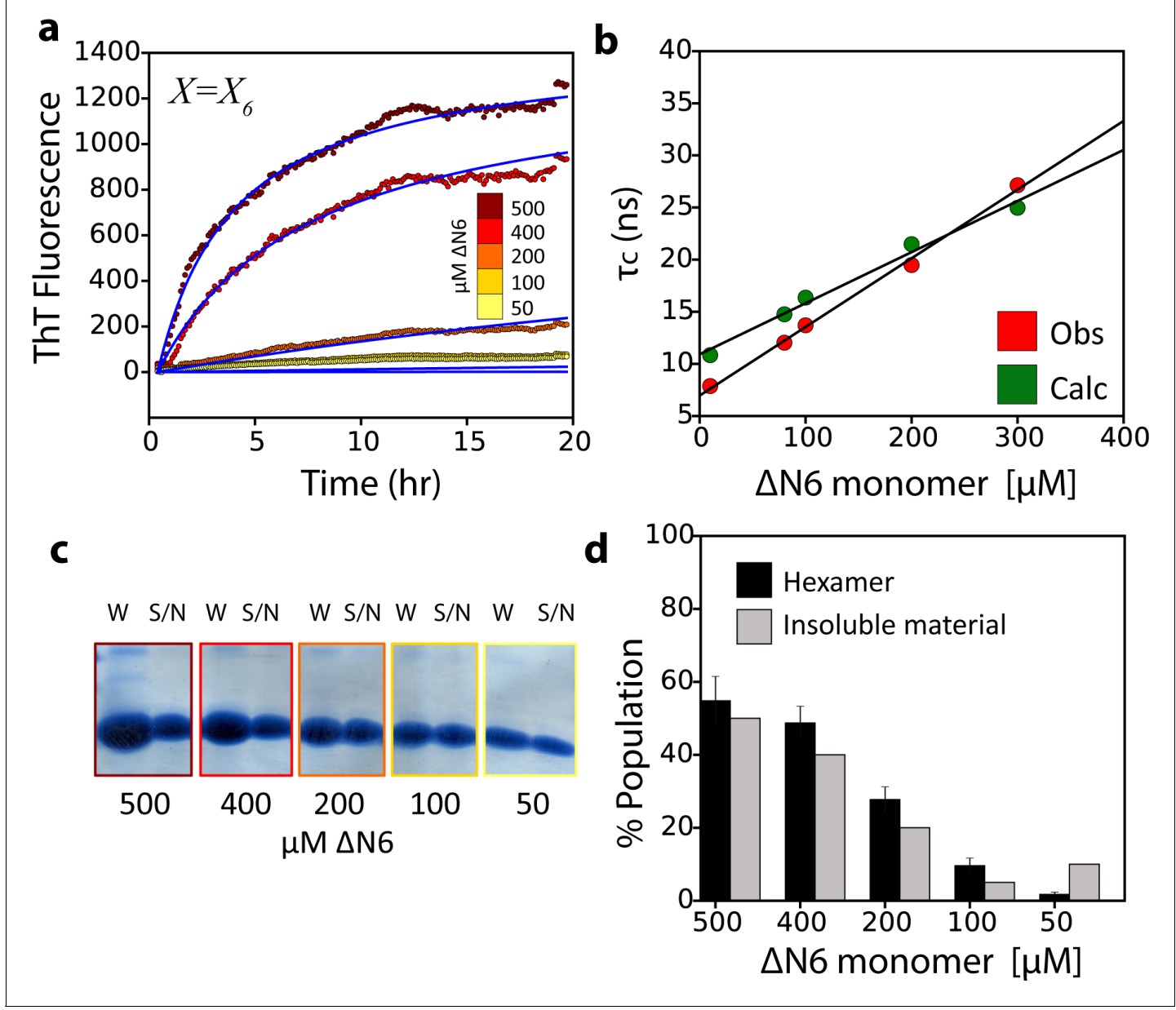

**Figure 7.** The monomer-dimer-hexamer model describes the thermodynamics and kinetics of fibril elongation. (a) Global fits (blue solid lines) to the fibril elongation kinetics monitored by ThT fluorescence assuming a hexamer addition model at different concentrations of soluble ΔN6 (dots) (Materials and methods and *Table 4*). The concentrations of ΔN6 are colored according to the key. The average of five replicates is shown. (b) Protein correlation times ($\tau_c$) measured using NMR (red) and back-calculated values (green) using the populations of monomers, dimers and hexamers predicted from the monomer-dimer-hexamer model and the correlation times of the dimers and hexamer structural models shown in *Figures 4* and *5*. (c) The fibril yield (after 100 hr) of each elongation reaction. SDS-PAGE analysis of the whole reaction (shown in (a)) before centrifugation (W) or of the supernatant (S/N) after centrifugation at the different concentrations of ΔN6, as indicated. (d) Bar-charts showing the % of insoluble material (gray) measured using densitometry of the gel shown in (c). The % hexamer population in the absence of seeds (black) predicted by the monomer-dimer-hexamer model at each ΔN6 concentration correlates with the % insoluble material (gray). Note that the fibril yield is low since fibrils cannot form when the monomer concentration falls significantly below the $K_d$ for dimer formation (50 μM).

DOI: https://doi.org/10.7554/eLife.46574.025

The following figure supplement is available for figure 7:

**Figure supplement 1.** Alternative kinetic models do not describe the kinetics of seeded fibril growth.
DOI: https://doi.org/10.7554/eLife.46574.026

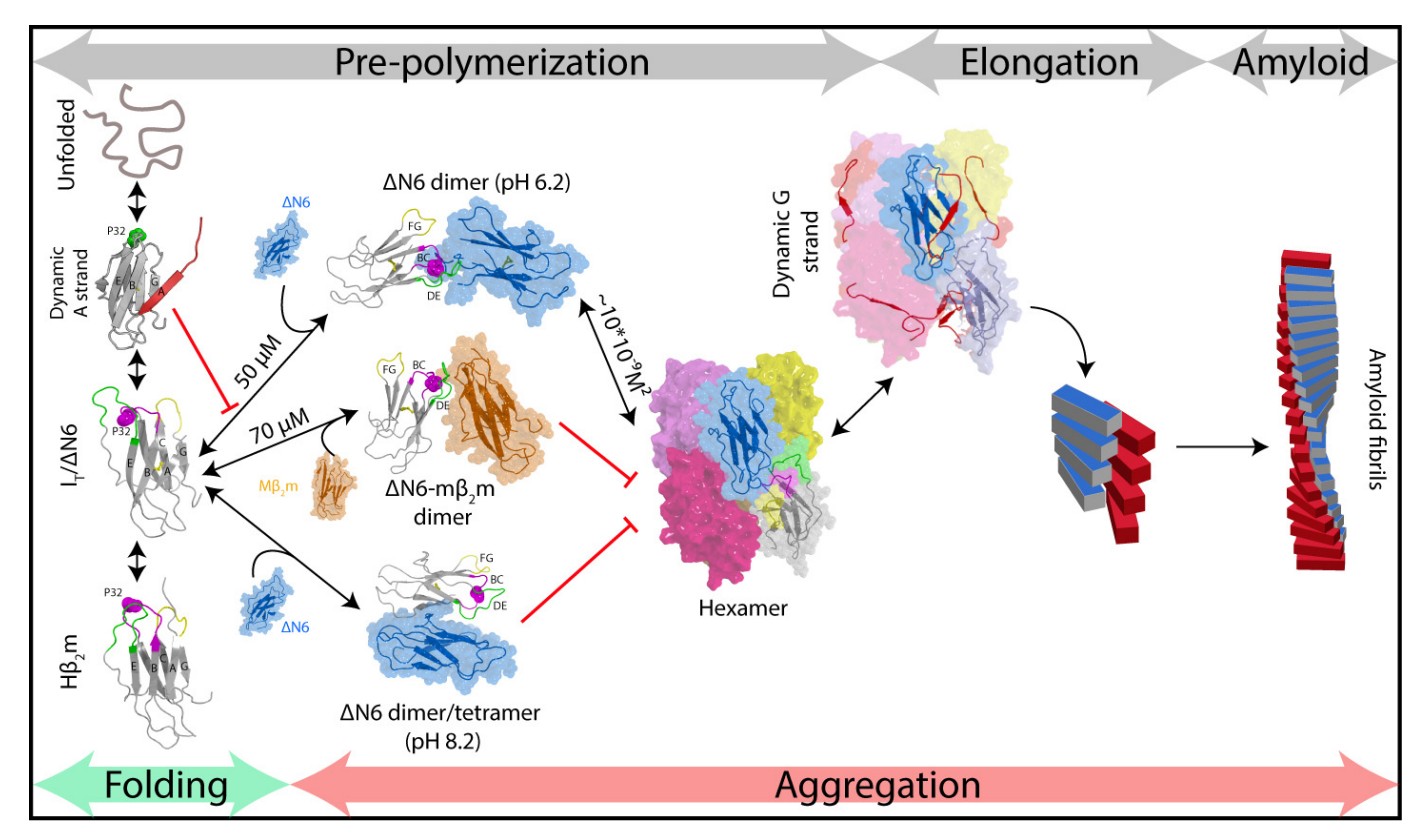

**Figure 8.** Fibril formation in atomic detail. Schematic representation of the mechanism of amyloid formation for ΔN6. During folding of hβ$_2$m, a highly dynamic intermediate with a flexible A strand is populated prior to formation of the native-like intermediate termed I$_T$, which has a native-like fold but contains a non-native *trans* X-Pro32 bond. The latter species is mimicked by ΔN6 and formed in vivo by proteolytic degradation of the WT protein (***Bellotti et al., 1998***). Only I$_T$/ΔN6 is primed for aggregation, while the intermediate with the flexible A strand is not able to assembly directly into amyloid (***Karamanos et al., 2016***). As reported here, ΔN6 forms elongated head-to-head dimers (upper image, center) which assemble into hexamers. Alternative dimers involving interactions between the ABED β-sheets in adjacent molecules formed at pH 8.2 (lower image, center) do not associate further into fibrils. Murine β$_2$m (mβ$_2$m) also interacts with ΔN6 at pH 6.2 to form head-to-head heterodimers. The subunit orientation is different in this heterodimer (***Karamanos et al., 2014***), occluding the hexamer interface and inhibiting assembly (central image). ΔN6 hexamers can elongate fibrillar seeds and show enhanced dynamics in the G strand which could represent the first step towards the major structural reorganization required to form the parallel in-register amyloid fold. How this final step occurs, however, remains to be solved.

DOI: https://doi.org/10.7554/eLife.46574.028

The following figure supplements are available for figure 8:

**Figure supplement 1.** A workflow to enable weakly self-assembling systems to be analyzed in structural, kinetic and thermodynamic detail.

DOI: https://doi.org/10.7554/eLife.46574.029

**Figure supplement 2.** Examples of some previously characterized oligomers of WT hβ$_2$m and ΔN6.

DOI: https://doi.org/10.7554/eLife.46574.030

## Chemical cross-linking and analytical SEC

ΔN6 (10 μM - 500 μM) was incubated at room temperature without shaking in 10 mM sodium phosphate buffer, pH 6.2 containing 83.3 mM NaCl (total ionic strength of 100 mM), 0.02% (*w/v*) NaN$_3$ overnight. A 100-fold molar excess of 1-ethyl-3-(3-dimethylaminopropyl)-carbodiimide hydrochloride (EDC) (final concentration 1 mM - 50 mM) was added to the reaction, incubated for 10 min with gentle mixing, followed by the addition of 5 mM sulpho-N-hydroxysuccinimide (NHS) for 5 min at room temperature. Cross-linking was then quenched by the addition of 10-fold molar excess (over the concentration of EDC) of Tris HCl pH 8.0, or for cellular cytotoxicity assays, Dulbecco's PBS, and samples were then analyzed immediately using an analytical Superdex S75 10/300 GL column (GE Healthcare) equilibrated with the same buffer. A similar protocol was used to cross-link ΔN6 during de novo fibril formation. A 500 μL volume of 80 μM ΔN6 in 10 mM sodium phosphate buffer, pH 6.2

containing 83.3 mM NaCl, 0.02% (w/v) NaN$_3$ was incubated in a 1.5 mL micro-centrifuge tube at 37°C with 600 rpm vigorous shaking on a thriller shaker. Under these conditions, the lag time is ~100 hr instead of ~20 hr when the protein is incubated in a 96-well plate (*Figure 2b* and *Figure 3—figure supplement 1a*). Samples were cross-linked at various time-points during assembly by addition of 8 mM EDC, incubated for 15 min, followed by addition of 5 mM NHS, also incubated at room temperature for 15 min. The cross-linking reaction was quenched by addition of 200 mM ammonium acetate before samples were subjected to analysis by SDS-PAGE. Given the unavoidable dilution of the samples and their re-equilibration during the SEC run, quantitative analysis of the SEC traces of cross-linked and uncross-linked samples was not performed.

## Measurement of ΔN6 correlation times

Rotational correlation times ($\tau_c$) of ΔN6 at different concentrations were measured in 10 mM sodium phosphate buffer pH 6.2 containing 83.3 mM NaCl, or the same buffer at pH 8.2 containing 86.6 mM NaCl (total ionic strength for each sample of 100 mM), 25°C using a $^1$H-TRACT experiment (*Lee et al., 2006*) with delays of 0.002–0.064 s in a Varian Inova NMR spectrometer operating at 750 MHz. At each delay, the signal intensity of resonances in the amide region was integrated and the resulting curve fitted to a single exponential decay function in order to calculate the relaxation rates of the TROSY ($R_\alpha$) and anti-TROSY ($R_\beta$) spins. The difference $R_\beta$ - $R_\alpha$ was then converted to the correlation time (*Lee et al., 2006*). Errors were calculated using duplicate measurements.

## Diffusion NMR measurements on ΔN6

Diffusion NMR experiments were performed on ΔN6 samples at different concentrations using pulsed field gradient (PFG) NMR spectroscopy using stimulated echoes with bipolar gradients performed on a Bruker Avance III 750 MHz spectrometer. A series of $^1$H spectra were collected as a function of gradient strength ($g$), the signal ($S$) was integrated and fitted to:

$$S/S_0 = \exp\left(-d \cdot g^2\right)$$

where $S_0$ is the signal intensity at zero field gradient, $d$ is the observed decay rate and $g$ is the strength of the field gradient pulses. The decay rate ($d$) is directly proportional to the diffusion coefficient, $D$, of the protein (*Stejskal and Tanner, 1965*).

## Chemical shift perturbation and calculation of K$_d$ values

$^1$H-$^{15}$N TROSY spectra of ΔN6 at different concentrations were collected using a 750 MHz Bruker Avance III spectrometer. The combined $^1$H and $^{15}$N chemical shift difference was calculated using the function:

$$\Delta\delta = \sqrt{\left(5 * \delta^1 H\right)^2 + \left(\delta^{15} N\right)^2}$$

Chemical shift data at 10 μM, 20 μM, 100 μM, 200 μM and 410 μM ΔN6 were fitted to a monomer (X$_1$) - dimer (X$_2$) – hexamer (X$_6$) model:

$$X_1 + X_1 \underset{k_1'}{\overset{k_1}{\rightleftarrows}} X_2 + X_2 + X_2 \underset{k_2'}{\overset{k_2}{\rightleftarrows}} X_6$$

The equilibrium concentration of hexamer [X$_6$] was calculated by numerical integration of the above model using scripts written in Python and converted to fractional saturation. The observed chemical shift ($\Delta\delta$) is then given by:

$$\Delta\delta = B_{max} \frac{6 * [X_6]}{[X_1]}$$

where $B_{max}$ is the maximum chemical shift difference. To obtain estimates for the monomer-monomer and dimer-hexamer K$_d$s a grid search was performed by fixing the dimer K$_d$ ($k'1/k1$) and the hexamer K$_d$ ($k'2/k2$) to different values (*Figure 2—figure supplement 2d*). Excellent fits were produced using a dimer K$_d$ <~50 μM, while the hexamer K$_d$ shows a narrow distribution centered at ~10 ± 5 x 10$^{-9}$ M (*Benilova et al., 2012*) (*Figure 2—figure supplement 2d*). To further validate

the estimation of the dimer $K_d$, RDC experiments were performed as a function of ΔN6 concentration (*Figure 2—figure supplement 2e*). ΔN6 was aligned in 10 mg/mL of PF1 phage (Asla Scientific) and $H_N$ RDCs were measured using ARTSY (*Fitzkee and Bax, 2010*). The biphasic behavior of the RDCs suggests a three-state equilibrium in agreement with the monomer-dimer-hexamer model. The first/second transition at lower protein concentration (blue/pink dashed line in *Figure 2—figure supplement 2e*) presumably reports on the monomer-dimer/dimer-hexamer equilibrium, respectively. In order to extract RDCs of the dimer species the blue dashed line was extrapolated to 100% dimer using various $K_d$ values. The resulting data were fitted to the structure of ΔN6 in order to calculate the alignment tensor of the dimer. Using a $K_d$ greater than 50 μM results in a decrease in the goodness of the fit (*Figure 2—figure supplement 2f*), unless a large conformational change in the monomer is invoked upon dimer formation. However, based on the chemical shift data shown in *Figure 2—figure supplement 2a,c* ΔN6 remains native-like across all concentrations, thus placing an upper limit of the dimer $K_d$ at ~50 μM in agreement with the grid search of the chemical shift data (*Figure 2—figure supplement 2d*). Note that the calculated tensor depends highly on the correct RDC values and therefore RDCs were not used in the structure calculations described below. Chemical shift perturbations for 10 residues that show significant chemical shift perturbations (11, 12, 23, 26, 50, 51, 52, 67, 68, 97) were fitted globally to this model, with representative examples shown in *Figure 2—figure supplement 2b*. Errors on the measured peak positions were calculated as the standard deviation of the mean for residues that show insignificant chemical shift changes. Errors on the fitted parameters were computed using Monte Carlo calculations with 100 steps.

To calculate populations of different species, a monomer-dimer-hexamer model was treated numerically, that is the kinetic equations that describe the time-evolution of the concentration of each species were integrated to $\tau=\infty$, after equilibrium was reached, yielding the equilibrium concentration (in molar units) of monomers, dimers and hexamers. Since the dimers consist of two monomers and hexamers of six monomers, these concentrations are then converted to populations (of monomers in the form of dimer or hexamer) using the relationship:

$$p_n = \frac{n[A_n]}{[M_{tot}]}$$

where n is the oligomer order, $[A_n]$ the equilibrium concentration of the oligomeric state and $[M_{tot}]$ the total protein concentration. The overall rate of assembly, $k_{on}^{over}$, for this three-state model is given by:

$$k_{on}^{over} = \frac{k_1^{app} k_2^{app}}{k_{-1} k_2^{app}},$$

where

$$k_1^{app} = 2k_1[M_{eq}]$$
$$k_2^{app} = 3k_2[D_{eq}]^2$$

And therefore:

$$k_{on}^{over} = \frac{k_1^{app} k_2^{app}}{k_{-1} k_2^{app}} = \frac{6k_1 k_2[M_{eq}][D_{eq}]^2}{k_{-1} + 3k_2[D_{eq}]^2}$$

The overall $k_{on}^{over}$ rate of assembly and therefore the total population of oligomers scales linearly as a function of the monomer concentration (see inset in Figure 2-Supplement 1f).

## PRE experiments

The ΔN6 variants ([14]N-labeled) S20C, S33C, L54C and S61C (1–2 mg/mL) were incubated with 5 mM DTT for 20 min, excess DTT was removed using a PD10 gravity column (GE Healthcare) and the protein was then labeled immediately with MTSL by incubation with a 40-fold molar excess (over the total ΔN6 concentration) of the spin label for 4 hr in 25 mM sodium phosphate buffer, pH 7.0 containing 1 mM EDTA at room temperature. Excess spin label was removed by gel filtration (PD10 column) in the same buffer. Spin-labeled ΔN6 was used directly or stored at −80˚C. In all cases, 100% labeling at a single site was confirmed using ESI-MS. For each PRE experiment, MTSL-labeled [14]N-Δ

N6 (10 μM −80 μM) was mixed with $^{15}$N-labeled ΔN6 (60 μM −240 μM) and the difference of the $^1$H R$_2$ rates between oxidized and reduced (the latter created by addition of 1 mM ascorbic acid) MTSL-labelled $^{14}$N-ΔN6 was measured. Experiments were performed in 10 mM sodium phosphate buffer, pH 6.2 containing 83.3 mM NaCl or 10 mM sodium phosphate buffer, pH 8.2 containing 86.6 mM NaCl (a total ionic strength of 100 mM at each pH value). Data were recorded at 25°C using a $^1$H-$^{15}$N correlation based pulse sequence with 5 or 6 time-points (0.0016–0.016 s) (*Clore and Iwahara, 2009*) and at least 32 scans per incremental delay, utilizing a Bruker Avance III 750MHz spectrometer equipped with a cryogenic probe. R$_2$ rates were extracted by fitting the relaxation data to single exponentials using in-house scripts. The H$_N$-Γ$_2$ rate was then calculated as the difference between the R$_2$ rate in the paramagnetic (R$_{2, para}$) versus diamagnetic (R$_{2, dia}$) sample:

$$\Gamma_2 = R_{2,para} - R_{2,dia}$$

Errors were calculated based on the noise of the experiment. The small PRE signal observed when ΔN6 is modified with MTSL at position 20 can be attributed to non-specific binding of the spin label itself to adjacent protein molecules, since addition of free MTSL resulted in a similar PRE profile (not shown). Thus, data arising from spin-labeled ΔN6 at position 20 were not included in quantitative analysis of the PRE experiments.

## $^{15}$N transverse relaxation dispersion CPMG experiments

$^{15}$N transverse relaxation dispersion CPMG experiments were performed as described in reference (*Loria et al., 1999*) using samples dissolved in 10 mM sodium phosphate buffer containing 83.3 or 86.6 mM NaCl to maintain a constant ionic strength of 100 mM at pH 6.2 and 8.2, respectively. Spectra were acquired using a Varian Inova 500 MHz spectrometer using a fixed relaxation delay (τ$_{cpmg}$) of 48 ms or a Bruker Avance III 750 MHz or 950 MHz spectrometer using a delay of 40 ms. Spectra were processed using NMRPipe (*Delaglio et al., 1995*) and $R_{2,eff}$ rates were calculated using:

$$R_{2,\,eff} = \frac{\left(\frac{I_x}{I_0}\right)}{\tau_{CPMG}}$$

where $I_x$ is the peak intensity in each experiment and $I_0$ is the peak intensity in the reference spectrum (with CPMG train applied). CPMG data from two clusters of residues, one reporting on intermolecular interactions (12 residues) and the second reporting on the dynamics of the G strand (four residues) (see text) were fitted globally to the Bloch-McConnell equations (*McConnell, 1958*) describing a two-state exchanging system using the software package 'relax' (*Morin et al., 2014*). The fact that dimer and hexamer interfaces partly overlap, complicates the analysis of the CPMG data at pH 6.2. However, at the concentrations used, where either hexamerization is low (180 μM: 26% monomer, 48% ΔN6 molecules as dimer, 26% ΔN6 molecules as hexamer) or dimerization remains constant (480 μM: 13% monomer, 32% ΔN6 molecules as dimer, 55% ΔN6 molecules as hexamer) good quality fits to a simple two-state model were obtained. The calculated exchange parameters report on both dimer and hexamer formation. Due to this ambiguity, the apparent exchange rates obtained by fitting the CPMG data were not used in the kinetic modeling of the reaction, but used instead to report on the apparent differences in exchange dynamics of different residues as hexamer formation is enhanced.

## Calculation of structural models

### Structural models of dimers

Simulated annealing calculations were carried out in XPLOR-NIH (*Schwieters et al., 2003*). To account for the flexibility of the MTSL moiety, the paramagnetic group was represented as a five-membered ensemble. The computational strategy employed included three PRE potential terms (arising from S61C-ΔN6, L54C-ΔN6 and S33C-ΔN6) and classic geometry restraints to restrict deviation from bond lengths, angles and dihedral angles. Resonances for which an estimation of the R$_2$ rate in the presence of the oxidized spin label was not possible were incorporated in the protocol as nOe-type restraints with an upper bound of 11.5 Å and a lower bound of 9 Å. Chemical shift perturbations observed upon binding were incorporated as sparse, highly ambiguous intermolecular

distance restraints as described in reference (*Clore and Schwieters, 2003*). As chemical shifts can be influenced by numerous factors upon protein-protein interaction, the treatment of the derived data undertaken here results in a loose potential term that is unlikely to bias the structure calculation. Finally, the protocol included a weak radius of gyration restraint ($R_{gyr}$) calculated as $2.2*N^{0.38}$, where N is the number of residues in the complex. $R_{gyr}$ is required in order to prevent bias towards more extended structures and tends to underestimate the true value of the radius of gyration (*Kuszewski et al., 1999*). C2 distance symmetry restraints alongside a non-crystallographic symmetry potential term were also implemented to ensure that the two monomers adopt the same conformation in the dimer. The aforementioned potential terms were then used in a rigid-body energy minimization/simulated annealing in torsion angle space protocol to minimize the difference between the observed and calculated $\Gamma_2$ rates, starting from random orientations. The first step in the structure calculation consisted of 5000 steps of energy minimization against only the sparse chemical shift restraints, followed by simulated annealing dynamics with all the potential terms active, where the temperature is slowly decreased (3000–25 K) over four fs. During the hot phase (T = 3000 K) the PRE and nOe terms were underweighted to allow the proteins to sample a large conformational space and they were geometrically increased during the cooling phase. Proteins were treated as rigid bodies until the initiation of the cooling phase, where side chains were allowed to float (semi-rigid body calculation). The final step included torsion angle minimization using all potential terms. The calculations converged to two dimer structures shown in *Figure 4a* (lowest energy, termed dimer A) and *Figure 4—figure supplement 1e* (dimer B). Both dimers show a head–head configuration with dimer B showing a larger interface which extends from the BC and DE loops to the B and E strands. Fits to the PRE data are of lower quality for dimer B as judged by visual inspection of the fits and the restraints violation (RMS) (*Table 1*). However, both dimers were used as initial building blocks for calculation of the hexamer models.

## Structural models of hexamers

Starting from dimer A or dimer B, an initial docking run was performed. Dimers were treated as rigid bodies and placed at random positions. Residues for which chemical shift differences were observed at high protein concentrations were used as sparse distance restraints alongside a geometry energy potential. Three-fold symmetry was imposed together with a non-crystallographic symmetry potential. The energy arising from the four potential energy terms was minimized in order to generate 1000 hexamer structures. The PRE potential energy was not used during the calculation but only in the scoring of the structures generated (together with the energy of the other four terms). Starting from dimer A, the plot of energy versus RMSD (to the lowest energy structure) (*Figure 5—figure supplement 3a*) shows the expected funnel shape with 44 of the 50 lowest energy structures sharing a backbone RMSD of 2–3 Å, indicating that these models are close to a structure that satisfies the PRE restraints. On the other hand, the 50 lowest energy hexamers built form dimer B show an RMSD of up to 35 Å with three clusters formed (*Figure 5—figure supplement 3b*). Therefore, these four hexamer structures (one arising from dimer A and three from dimer B) were taken forward for the next round of the protocol which consisted of an exhaustive simulated annealing calculation. Since it is difficult to define the extent to which the PREs arise from the dimer and hexamer, the PREs restraints were converted to distance restraints. Residues that show high PREs such that no peak was observed in the spectrum with oxidized MTSL were given no lower bound, while residues not affected by MTSL had no upper bound. This strategy removed some of the dimer – hexamer ambiguity and instead the protocol searched for hexamers that generally interact in the areas which show increased $\Gamma_2$ rates at high $\Delta$N6 concentrations, rather than quantitatively fitting the PRE data. The details of the simulated annealing run were similar to that performed to calculate the dimer structure, but included a three-fold (instead of two-fold) distance symmetry potential term (giving rise to hexamers with a D3 overall symmetry). The final stage of the protocol consisted of refinement in explicit water using XPLOR-NIH. Distances were converted back to PREs to allow comparison with the measured PRE data. Following this protocol, the hexamers produced from dimer A show increased PRE rates in the A strand and BC, DE loops in agreement with the PRE data (*Figure 5—figure supplement 4*). On the other hand, all hexamers assembled from dimer B show calculated PREs which describe the measured PREs less well (*Table 2*) (these fits are available on the University of Leeds publicly available library [https://doi.org/10.5518/329]). Note that dimer and hexamer

models were generated and selected based only on the agreement with the NMR data. Cross-sections of the oligomers obtained from other experiments were used only as a check of consistency with the models determined. PDBs of the dimers and hexamers have been deposited in the University of Leeds publicly available library (https://doi.org/10.5518/329). The buried surface areas of dimers and hexamers were calculated using the program NACCESS (*Hubbard and Thornton, 1993*) which calculates the per residue accessible surface area (ASA) given a structural model. A cutoff of 10% loss in ASA between monomers and dimers/hexamers was used.

## Native ESI-IMS-MS

ΔN6 samples were exchanged into a buffer consisting of 50 mM ammonium acetate, 50 mM ammonium bicarbonate pH 7.4 using Zeba spin desalting columns (Thermo Scientific) immediately before MS analysis. NanoESI–IMS–MS spectra were acquired using a Synapt HDMS mass spectrometer (Waters) with platinum/gold-plated borosilicate capillaries prepared in house. Typical instrument parameters were: capillary voltage, 1.2–1.6 kV; cone voltage, 40 V; trap collision voltage, 6 V; transfer collision voltage, 10 V; trap DC bias, 20 V; backing pressure, 4.5 mbar; IMS gas pressure, 0.5 mbar; traveling wave height, 7 V; and traveling wave velocity, 250 ms$^{-1}$. Data were processed with MassLynx v4.1 and Driftscope 2.5 (Waters). Collison cross sections (CCSs) were estimated through a calibration approach using arrival-time data for ions with known CCS values (β-lactoglobulin A, avidin, concanavilin A and yeast alcohol dehydrogenase, all from Sigma Aldrich). Estimated modal CCSs are shown. Theoretical CCSs were calculated for hexameric model structures using the scaled projection approximation method implemented in IMPACT (*Marklund et al., 2015*) after performing *in vacuo* molecular dynamics simulations to account for structural alterations arising from transfer into the gas-phase, as previously described (*Devine et al., 2017*). Note that the best scoring model agrees with the CCS of the lowest charge state (15+) (which is considered to be most native; *Vahidi et al., 2013*) of the hexamer derived independently using the NMR data alone. The IMS-MS experiments thus serve as an independent validation of the structural model derived.

## ANS binding

The ability of different ΔN6 species to bind 8-anilinonaphthalene-1-sulfonic acid (ANS) was measured by mixing 50 µL of each fraction obtained from analytical SEC of 1 µM ΔN6 (see above) with 200 µL of ANS to yield a final concentration of ANS of 200 µM. Fluorescence spectra were recorded using a ClarioStar plate reader (BMG Labtech) using an excitation wavelength of 370 nm and emission from 400 to 600 nm. The concentration of protein used was estimated to be ~240 µM (monomer), 3 µM (dimer) and 1 µM (hexamer). Experiments on Im7 L53A I54A were performed as described in reference (*Spence et al., 2004*).

## Cytotoxicity assays

ΔN6 (240 µM) was cross-linked with EDC/NHS as described above. 500 µL of cross-linked material was resolved using a Superdex 75 analytical gel filtration column (GE Healthcare) using Dulbecco's PBS as a mobile phase (Sigma #D8537). 1 mL fractions were collected. SH-SY5Y cells were obtained from an authenticated and mycoplasma free source (ATCC CRL-2266) and were passaged up to 10 times. The cells were mycoplasma tested and found to be negative. The cells were cultured as described previously (*Xue et al., 2009*) using 15,000 cells per well in 96-well plates (Corning #3595) for 24 hr in 100 µl of growth medium. This time point has been widely used in other studies of cytotoxicity and hence allows comparison of the results obtained with observations on β$_2$m and other amyloid systems (*Fusco et al., 2017*; *Campioni et al., 2010*; *Xue et al., 2009*; *Leri et al., 2016*; *Giorgetti et al., 2008*).

Cells were then incubated with 50 µL of each fraction from SEC for 24 hr before analyzing cell viability. PBS alone was used as negative control and 0.02% (*w/v*) NaN$_3$ was added as a positive control for cell death. Each experiment consisted of at least three repeats from two independent cross-linking reactions. The neuroblastoma cell line SH-SY5Y was chosen for our assays, as this cell line is a widely accepted model for the study of amyloid toxicity and has been used by other laboratories for β$_2$m and other amyloid-forming sequences (*Fusco et al., 2017*; *Campioni et al., 2010*; *Xue et al., 2009*; *Leri et al., 2016*; *Giorgetti et al., 2008*).

For MTT assays, 10 µL of a 10 mg/mL solution of MTT (Sigma-Aldrich) was added to each well for 1.5 hr. Cell growth media and excess MTT were then removed and reduced MTT was solubilized using 50 µL DMSO per well. The absorbance of MTT was determined using a ClarioStar plate reader (BMG Labtech) at 570 nm with background subtraction at 650 nm. MTT reduction was calculated as a percentage of PBS buffer treated controls (100%) and cells treated with 0.02% (w/v) $NaN_3$ (0%).

Cellular ATP was measured using the ATPlite Luminescence ATP detection assay (#6016963 Perkin Elmer) according to the manufacturer's protocol. Luminescence was measured on a PolarStar OPTIMA plate reader (BMG Labtech). Cellular ATP was calculated as a percentage of PBS-buffer-treated controls (100%) and cells treated with 0.02% (w/v) $NaN_3$ (0%).

Lactate dehydrogenase (LDH) release was measured using a Pierce LDH cytotoxicity assay kit (#88953 ThermoFisher Scientific) according to the manufacturer's instructions. Absorbance was determined using a ClarioStar plate reader (BMG Labtech) at 490 nm with background subtraction at 680 nm. LDH release was calculated and normalized to detergent lysed cells (100%) and PBS-buffer-treated controls (0%).

Reactive oxygen species (ROS) production was determined using 10 µM 2',7'-dichlorohydrofluorescein diacetate ($H_2$DCFDA) (#D399 ThermoFisher Scientific). Cells were incubated with $H_2$DCFDA for 10 min prior to the addition of ΔN6 samples from SEC. Fluorescence was recorded after further incubation for 45 min using a ClarioStar plate reader (BMG Labtech) at 540 nm. ROS production was calculated as a percentage of PBS buffer treated controls (100%) and cells treated with 0.02% (w/v) $NaN_3$ (0%). 10 µM $H_2O_2$ was used as a positive control for the induction of ROS production and resulted in a 373 ± 21% ROS assay signal compared with incubation with PBS. Each experiment consisted typically of two-to-three independent experiments each containing five replicates per condition. The error bars represent mean S.E, * p 0.05. Raw data are available at (https://doi.org/10.5518/329).

## Kinetic modeling of the rates of amyloid formation

The fibril growth kinetics for ΔN6 in the presence of ΔN6 fibril seeds shown in *Figure 1c,d* were fitted to five different kinetic models which consisted of two distinct modules (pre-polymerization and polymerization). In model (1) monomers are assumed to add to the fibril ends (this model contains two parameters, the elongation rate, $k_e$, and a fibril depolymerization rate, $k_e'$). In model (2) the monomers are assumed to be in conformational exchange with a monomeric excited state that is responsible for elongation. Model (3) includes a monomer-dimer equilibrium followed by dimer addition to the fibril ends. Models (2) and (3) contain four parameters, monomer-monomer binding/unbinding rates ($k_1$ and $k_1'$) and monomer conformational exchange rates ($k_e$, $k_e'$). In the fourth model (4) a monomer-dimer-tetramer-hexamer equilibrium was considered. Finally, in model (5) a monomer-dimer-hexamer equilibrium was considered. This model contains six parameters, $k_1$, $k_1'$ (monomer-monomer binding), $k_2$, $k_2'$ (dimer-dimer-dimer binding) and $k_e$, $k_e'$ (exchange). The rate equations for all models are listed in *Table 4* and were solved numerically using in house scripts written in Python. In the polymerization module, that describes the addition of the elongation unit (X) to the already formed fibrils, each assembly step was modeled explicitly (*Table 4*). The primary output of each model is the mass fraction of each species as a function of time. To convert the output of the program to ThT fluorescence curves, the mass of the elongated seeds was multiplied by a fluorescence factor ($K_{tht}$). Elongated seeds were assumed to be any species ($F_i$) that contain more monomers than the preformed seeds added in the assay ($F_0$) ($1 \leq i \leq N$), where N represents the number of monomers in the fibril at the end of the reaction. The mass fraction of monomers present in a fibril was assumed to scale linearly with ThT fluorescence, giving the following expression for calculating the progress curves:

$$\mathbf{F_i(t)} = \sum_{i=1}^{N} \mathbf{i[F_i]K_{tht}}$$

Seeding data using all five starting ΔN6 monomer concentrations were fitted to each model globally sharing all rate constants using N = 200 (200 monomers in a fibril which would correspond to a fibril roughly 500 nm in length; *White et al., 2009*). The monomer-dimer $K_d$ value ($k_1'/k_1$) was fixed to 50 µM. Fitting the kinetic data to the hexamer addition model produces a hexamer $K_d$ of ~1.9×$10^{-9}$ $M^2$ similar to the value of ~10 ± 5 x $10^{-9}$ M (*Benilova et al., 2012*) $K_d$ obtained by

fitting the chemical shift perturbation data on protein concentration, confirming the robustness of the model and the approach employed. Using the estimated $K_d$ values to obtain the populations of dimer and hexamer ($P_{dim}$, $P_{hex}$) and the structural models to calculate correlation times of the dimers and hexamers ($\tau_{c,dim}$, $\tau_{c,hex}$), the apparent correlation time at each $\Delta N6$ concentration ($\tau_{c,app}$) (computed as $\tau_{c,app} = P_{mon} \tau_{c,mon} + P_{dim} \tau_{c,dim} + P_{hex} \tau_{c,hex}$, where $P_{mon/dim/hex}$ is the population of dimer/hexamer, respectively and $\tau_{c,mon/dim/hex}$ is the correlation time of each species (9.8, 18.5 and 60.3 ns, respectively) calculated using the structural models by HYDROPRO (*Ortega et al., 2011*) matches the NMR measured $\tau_c$ versus $\Delta N6$ concentration (*Figure 7b*).

## Acknowledgements

We thank members of the Radford laboratory for helpful discussions, Nasir Khan for his excellent technical support and Amy Barker for her assistance with AUC. We also thank Alison Ashcroft for her long-term collaboration on native MS. TKK, SCG, EEC, EWH and SER acknowledge funding from the Wellcome Trust (089311/Z/09/Z, 204963 and 109154/Z/15/Z) and the European Research Council (ERC) under European Union's Seventh Framework Programme (FP7/2007-2013) ERC grant agreement no. 322408. ANC is funded by the BBSRC (BB/K000659/1). We acknowledge the Wellcome Trust (094232) and the University of Leeds for funding the NMR instrumentation and the BBSRC (BB/E012558/1) for providing funds for the Synapt HDMS mass spectrometer.

## Additional information

### Funding

| Funder | Grant reference number | Author |
|---|---|---|
| Wellcome Trust | 089311/Z/09/Z | Theodoros K Karamanos<br>Sheena E Radford |
| Wellcome Trust | 204963 | Sheena E Radford |
| Wellcome Trust | 109154/Z/15/Z | Emma E Cawood<br>Sheena E Radford |
| European Research Council | 322408 | Theodoros K Karamanos<br>Matthew P Jackson<br>Sheena E Radford |
| Biotechnology and Biological Sciences Research Council | BB/K000659/1 | Antonio N Calabrese<br>Sheena E Radford |
| Wellcome Trust | 094232 | Arnout P Kalverda<br>Sheena E Radford |
| Biotechnology and Biological Sciences Research Council | BB/E012558/1 | Sheena E Radford |
| Wellcome Trust | 092896MA | Theodoros K Karamanos<br>Sophia C Goodchild<br>Sheena E Radford<br>Eric W Hewitt |

The funders had no role in study design, data collection and interpretation, or the decision to submit the work for publication.

### Author contributions

Theodoros K Karamanos, Matthew P Jackson, Antonio N Calabrese, Conceptualization, Data curation, Formal analysis, Investigation, Methodology, Writing—original draft, Writing—review and editing; Sophia C Goodchild, Conceptualization, Data curation, Formal analysis, Investigation, Writing—review and editing; Emma E Cawood, Conceptualization, Investigation, Writing—review and editing; Gary S Thompson, Conceptualization, Formal analysis, Investigation, Writing—review and editing; Arnout P Kalverda, Formal analysis, Methodology, Writing—review and editing; Eric W Hewitt, Conceptualization, Supervision, Investigation, Methodology, Writing—original draft, Project

administration, Writing—review and editing; Sheena E Radford, Conceptualization, Supervision, Funding acquisition, Writing—original draft, Project administration, Writing—review and editing

## Author ORCIDs

Theodoros K Karamanos (iD) https://orcid.org/0000-0003-2297-540X
Antonio N Calabrese (iD) http://orcid.org/0000-0003-2437-7761
Emma E Cawood (iD) http://orcid.org/0000-0002-2707-8022
Eric W Hewitt (iD) http://orcid.org/0000-0002-6238-6303
Sheena E Radford (iD) https://orcid.org/0000-0002-3079-8039

## Decision letter and Author response

Decision letter https://doi.org/10.7554/eLife.46574.039
Author response https://doi.org/10.7554/eLife.46574.040

# Additional files

## Supplementary files

• Transparent reporting form
DOI: https://doi.org/10.7554/eLife.46574.031

## Data availability

Data are freely available at the University of Leeds Data Depository: https://doi.org/10.5518/329.

The following dataset was generated:

| Author(s) | Year | Dataset title | Dataset URL | Database and Identifier |
|---|---|---|---|---|
| Karamanos TK, Jackson MP, Calabrese AN, Good-child SC, Cawood EE, Thompson GS, Kalverda AP, Hewitt EW, Radford SE | 2019 | Data from: Structural mapping of oligomeric intermediates in an amyloid assembly pathway | https://doi.org/10.5518/329 | University of Leeds Data Depository, 10.5518/329 |

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
