## [Decision Letter]

Thank you for submitting your article "Structural mapping of oligomeric intermediates in an amyloid assembly pathway" for consideration by *eLife*. Your article has been reviewed by two peer reviewers, and the evaluation has been overseen by a Reviewing Editor and John Kuriyan as the Senior Editor. The reviewers have opted to remain anonymous.

The reviewers have discussed the reviews with one another and the Reviewing Editor has drafted this decision to help you prepare a revised submission.

Summary:

This manuscript from Radford and colleagues addresses the pathway of amyloid assembly by β_2_-microglobulin. The studies focus on characterizing both the structure and physiological impact of the assembly-competent species, using changes in pH and a truncated form of the protein to gain these insights. In particular, extensive experimental and modelling analyses are combined to gain unique insights into the early stages of amyloid formation for this protein, which has been found in amyloid deposits in patients with dialysis-related amyloidosis. The results provide an atomic view of ΔN6 retaining a native-like structure upon forming dimers and hexamers, which do not exhibit toxicity. This finding is noteworthy since oligomers have often been reported to be particularly cytotoxic, and the structure is distinct from other previously studied forms of the protein. The new results here help explain previous studies and provide clues as to how the major structural conversion to amyloid may occur.

Essential revisions:

The reviewers have identified a number of key questions, which should be addressed in a revised manuscript. The reviewers raised multiple questions regarding the conclusions and have requested additional information on the identity, quantitation, and thermodynamics to support them:

Hexamer abundance and structure:

1) The experimental determination of the hexamer K_d_ at ~10 ± 5 nM^2^ (ranging to 1.9 nM^2^ for kinetic modelling) is not consistent with the data. For example, using the values reported in the paper for 180 µM monomer of 48% dimer and 24% hexamer (subsection “Structural models of on-pathway hexamers”, first paragraph), assuming the remainder, 28%, is monomer, K_d_ for dimer to monomer is 59 µM and K_d_ for hexamer to dimer is 11x10^-9^ M^2^ = 11 x 10^9^ nM^2^ = 11 x 10^3^ µM^2^. This also will impact the derived ΔGº value. Use of the incorrect value for the hexamer K_d_ might have major consequences throughout the paper, though probably the conclusions will for the most part be unaltered. Elsewhere in the paper, for 180 µM monomer the% dimer is 36% and% hexamer is 15, 14 or 17%. There are also variations in% dimer and hexamer for 480 µM, etc. The authors should address the reasons for the variations.

2) The AUC seems to show that at low protein concentrations (where there presumably should be very few of these), there is equal amount of monomer and hexamer protein if not more hexamer. It's unclear how these qualify as rarely populated states, so further or a more quantitative explanation would help clarify this.

3) Subsection “Kinetic modelling of the rates of amyloid formation”, and correlation time calculations for Figure 7. Monomer will be present (particularly at the lower concentrations), why is this not included in the calculation? The omission of monomer may explain why the calculated values are too high at low protein concentration. Please also provide more specifics on how the correlation time was calculated for the dimer and hexamer from the models of the structures. It may be of interest to include tetramer in Figure 7—figure supplement 1.

4) Figure 2—figure supplement 1 – why are the signal intensities so different in panels A and B when nominally the same amount of protein was loaded onto the same column? It seems likely that the very small higher order oligomer signal that is seen in panel B is also present in panel A if the signal magnitudes were comparable. It seems important to address this since establishing the nature (i.e. dissociating upon dilution in the absence of cross-linking) and size of the excited species is important to the conclusions of the manuscript

5) It would be of interest to further define the hexamer and distinguish between the possible structural models. For Figure 6, please provide additional specifics for the analysis of protein concentration dependence of relaxation e.g. define the k_ex_ values in terms of constituent processes/terms and the observed protein concentration dependences. Can different protein concentration dependence for different residues be used to distinguish the dimer and hexamer interfaces? For the process involving strand G, is the chemical shift change towards random coil? Could strand G be involved in an interface? Also, for data in Figure 5 and its figure supplements, can more information be determined e.g. Figure 5—figure supplement 1 could include additional PRE data; Figure 5—figure supplement 2, consideration of residues 63, 64; Figure 5—figure supplement 4, why is the back calculation for hexamer model A done for PRE data at 120 µM where hexamer will be minimally formed? The agreement is stated to be good, but this is not clear.

6) Figure 5—figure supplement 5: Based on all 3 charge states, multiple hexamer models appear possible, however, the 15^+^ state is the main one considered. Please clarify the rationale for this analysis. The Materials and methods describe ANS measurements for dimer, but no dimer data are shown. While not essential, it would be of interest to include such data for the dimer, and this may provide support for the modelled hydrophobic surface in the dimer. Please clarify the extent of fluorescence change when ANS binds to the Im7. The presentation is a bit confusing because hexamers show a larger increase ANS than monomer, but hexamers are stated to show only a small increase.

Dynamics:

1) Please address when ΔN6 samples may or may not be at equilibrium in the various experiments, and how this may impact the data analysis.

2) Subsection “A unified model of ∆N6 polymerization”: how does Figure 2B shows disappearance of hexamers during the elongation phase?

3) It appears that relatively little monomer is consumed in the aggregation reactions (for example, consider the gel in Figure 2B). This is not addressed directly in the paper and it seems like having the majority of the material not aggregation prone must have a large effect on the interpretation of the data, i.e. the aggregation assays are reporting not on the major process that is occurring in the solution, but a relatively minor side reaction. There is not a gel for the pH 2 aggregation experiments, but it seems likely that must more of the monomer material is consumed?

4) Hexamer population is reported to correlate with amount of aggregated protein; please elaborate on the possible basis for this correlation.

Cell Toxicity:

The cell toxicity studies are weak and should be removed; a single concentration at a single timepoint in a neuroblastoma cell line is that is of questionable physiological relevance is considered.

[Editors' note: further revisions were requested prior to acceptance, as described below.]

Thank you for resubmitting your work entitled "Structural mapping of oligomeric intermediates in an amyloid assembly pathway" for further consideration at *eLife*. Your revised article has been favorably evaluated by John Kuriyan as the Senior Editor, and Tricia Serio as the Reviewing Editor.

The manuscript has been improved but there are some remaining issues that need to be addressed before acceptance, as outlined below. Note that no further experimentation is required to address these issues.

1) Questions regarding quantitative analysis remain:

a) The proportions of monomer, dimer and hexamer for 120 μM total monomer are not consistent with those at higher concentrations (for 120 μM, the K_d_ dimer is 57 μM versus 50 μM, this should be addressed). The additional information on how the relative populations and K_d_s were calculated is pertinent, but not clear. Please define konover in the paper, give information on the origin of this equation (including reference), and include equations for how k_on_ overall is related to _τ_c and d.

b) Some additional support is needed for the authors' conclusions regarding larger species not really being quantitatively observable for AUC, but also SEC experiments, where the absorbance for larger species appears lower than expected. Can the authors use absorbance values more quantitatively?

c) Additional information is needed to show that independently prepared samples may indeed be compared i.e. they do not differ in uncontrolled ways due to unaccounted for variations in the timecourses or levels of assembly (oligomerization/aggregation), i.e. can the samples be considered to be consistently comparable (prior to and after completion of the time course of assembly/aggregation)?

d) Figure 2B: hexamer appears at 48 hours and disappears, while monomer appears unchanged. Presumably this is because at 80 μM total monomer, relatively little hexamer forms and gives rise to aggregates. So, one may reach an equilibrium at ~100 hours, where only monomer and dimer are significantly populated. It may be helpful to add some text or a figure to make this point clear, earlier in the paper. As for the lack of observation of dimer, could this be because the dimer is not effectively crosslinkable?

e) Given the experimental considerations the normalization of the chromatography signals may be reasonable; however, the relative signals within each panel are not internally consistent. Why is this?

2) Questions remain regarding the toxicity assay:

a) The toxicity assays are at best only a weakly supporting data: it is unclear why this is at all relevant for β_2_-microglobulin which does not specifically impact neuronal cells. It seems likely the cells simply don't take up β_2_-microglobulin, and good positive controls for these experiments are lacking.

---

## [Author Response]

Essential revisions:The reviewers have identified a number of key questions, which should be addressed in a revised manuscript. The reviewers raised multiple questions regarding the conclusions and have requested additional information on the identity, quantitation, and thermodynamics to support them:Hexamer abundance and structure:1) The experimental determination of the hexamer K_d_ at ~10 ± 5 nM^2^ (ranging to 1.9 nM^2^ for kinetic modelling) is not consistent with the data. For example, using the values reported in the paper for 180 µM monomer of 48% dimer and 24% hexamer (subsection “Structural models of on-pathway hexamers”, first paragraph), assuming the remainder, 28%, is monomer, K_d_ for dimer to monomer is 59 µM and K_d_ for hexamer to dimer is 11x10^-9^ M^2^ = 11 x 10^9^ nM^2^ = 11 x 10^3^ µM^2^. This also will impact the derived ΔGº value. Use of the incorrect value for the hexamer K_d_ might have major consequences throughout the paper, though probably the conclusions will for the most part be unaltered. Elsewhere in the paper, for 180 µM monomer the% dimer is 36% and% hexamer is 15, 14 or 17%. There are also variations in% dimer and hexamer for 480 µM, etc. The authors should address the reasons for the variations.

We thank the reviewers for pointing out the discrepancies in the quoted populations in our manuscript. These have been corrected throughout. We apologise that we did not spot this in our submitted manuscript.

We also thank the reviewer for pointing out that we made an error in converting the hexamer K_d_ from M^2^ to nM^2^. We agree with the reviewer that the hexamer K_d_ is 11 x 10^-9^ M^2^ = 11 x 10^9^ nM^2^ = 11 x 10^3^ µM^2^, as can be calculated from the% monomer, dimer and hexamer at 180 µM of 26%, 48% and 26%, respectively. It is important that we point out here that all calculations in our submitted manuscript were performed using the correct populations and K_d_. of 11 x 10^-9^ M^2^. Our error was in converting the units of M^2^ to nM^2^, which was only done editorially when writing the manuscript. Hence the error has no effect on any of the results presented. We apologise for making this error and the concerns it resulted in. The ΔGº value for hexamer formation accordingly is ~ 4 kJ/mol. We have corrected these values in the revised version of our manuscript.

In re-reading our manuscript, we also realised that we were not clear in describing how the relative populations and K_d_s were calculated. In our revision (revised Materials and methods subsection “Chemical shift perturbation and calculation of K_d_ values”), we now explain in detail how this was performed.

2) The AUC seems to show that at low protein concentrations (where there presumably should be very few of these), there is equal amount of monomer and hexamer protein if not more hexamer. It's unclear how these qualify as rarely populated states, so further or a more quantitative explanation would help clarify this.

In a system of interacting species (such as ΔΝ6) the measured S value obtained by sedimentation velocity AUC is a population-weight average. Thus peak areas cannot be interpreted directly as populations. We have clear evidence for hexamer formation from cross-linking experiments both during assembly (SDS PAGE) (Figure 2B) and at equilibrium (via SEC) (Figure 2—figure supplement 1). While the sedimentation velocity data are consistent with hexamers we cannot rule out the possibility that the peak observed corresponds to a weight average of species, some of which may be larger than hexamer. We have amended the subsection “Native-like dimers and hexamers form during ΔN6 assembly” to make this point clear.

3) Subsection “Kinetic modelling of the rates of amyloid formation”, and correlation time calculations for Figure 7. Monomer will be present (particularly at the lower concentrations), why is this not included in the calculation? The omission of monomer may explain why the calculated values are too high at low protein concentration.

We thank the reviewers for pointing this out. Monomers were indeed taken into account in the calculation, and we erroneously omitted to state this in the Materials and methods. We apologise. We have now corrected this in the revised version (subsection “Kinetic modelling of the rates of amyloid formation”).

Please also provide more specifics on how the correlation time was calculated for the dimer and hexamer from the models of the structures.

The correlation times were calculated using the structural models of the monomers, dimers and hexamers using HYDROPRO. We have added these details into the Materials and methods subsection “Kinetic modelling of the rates of amyloid formation”.

It may be of interest to include tetramer in Figure 7—figure supplement 1.

We have now included this figure in our revised manuscript. The addition of two additional rate constants (for dimer-tetramer formation) does not produce a significant improvement in the quality of the fits (new Figure 7—figure supplement 1D) compared with the monomer-dimer-hexamer model shown in Figure 7A. We therefore, chose to focus on the simpler model of monomer-dimer-hexamer.

4) Figure 2—figure supplement 1 – why are the signal intensities so different in panels A and B when nominally the same amount of protein was loaded onto the same column? It seems likely that the very small higher order oligomer signal that is seen in panel B is also present in panel A if the signal magnitudes were comparable. It seems important to address this since establishing the nature (i.e. dissociating upon dilution in the absence of cross-linking) and size of the excited species is important to the conclusions of the manuscript

The SEC experiments on crosslinked and uncrosslinked material was performed on different days and on different AKTA instruments using different sample loops and volumes loaded. Therefore, the absolute signal intensity between panels a and b cannot be compared directly. However, the curves *within each panel* can be compared, and show that without crosslinking (panel a) only monomer and dimers are present in contrast to after cross-linking (panel b). To avoid this confusion, we have now normalised the signal in each plot to the maximum intensity in each panel.

5) It would be of interest to further define the hexamer and distinguish between the possible structural models.

In the course of our work we also evaluated hexamer models assembled from dimer B in Figure 5—figure supplement 3 (see Author response images 1-3). Importantly, hexamer models assembled from dimer B predict large PREs in regions of the protein other than the BC, DE and FG loops which is inconsistent with the observed data. For example, hexamer i, (Author response image 1) shows large PREs in the F strand an succeeding FG loop, hexamer ii (Author response image 2) shows very large PREs from residue 33 to the N-terminus and the FG, while hexamer iii (Author response image 3) shows PREs from S61C to the C strand. We chose not to show all of these structure calculations in the supplement for the sake of brevity and clarity. We have made these data available on the University of Leeds publicly available library (https://doi.org/10.5518/329) and noted this in our revised manuscript (subsection “Structural models of hexamers”).

**Author response image 1. respfig1:** Evaluation of hexamer i, assembled for dimer B. Back-calculated PRE rates (black lines) are shown in panels A-D. Top (**E**) and side (**F**) view of the structural model.

**Author response image 2. respfig2:** Evaluation of hexamer ii, assembled for dimer B. Back-calculated PRE rates (black lines) are shown in panels A-D. Top (**E**) and side (**F**) view of the structural model.

**Author response image 3. respfig3:** Evaluation of hexamer iii, assembled for dimer B. Back-calculated PRE rates (black lines) are shown in panels A-D. Top (**E**) and side (**F**) view of the structural model.

For Figure 6, please provide additional specifics for the analysis of protein concentration dependence of relaxation e.g. define the k_ex_ values in terms of constituent processes/terms and the observed protein concentration dependences. Can different protein concentration dependence for different residues be used to distinguish the dimer and hexamer interfaces?

Unfortunately, the complexity of the monomer-dimer-hexamer model, combined with the limited range of protein concentrations that could be used to perform the experiments (as the protein aggregates non-specifically at higher concentrations than those described in the manuscript), mean that we do not have sufficient data to enable us to interpret the CPMG experiments in the detail suggested. To do so would be a substantial undertaking, beyond the scope of the current paper.

For the process involving strand G, is the chemical shift change towards random coil? Could strand G be involved in an interface?

CPMG experiments do not provide the sign of the Δδ (but just its absolute per-residue value), and so we cannot say whether the peaks move towards their random coil values. We think it is unlikely that the G strand is in an interface, since just four residues, all localised in the same part of the protein, show this behaviour. This point is now made in the subsection “Hexamer dynamics may prime further assembly into amyloid”. Most importantly, no significant concentration-dependent chemical shifts are observed for these four residues, nor are any PREs are observed for these residues at low or high protein concentration. A k_ex_ that is close to an order of magnitude slower for an interface involving the G strand compared with the interface at the apical part of the protein is also unlikely. Hence, we conclude that it is unlikely that the G strand is involved in an interface (see the aforementioned subsection).

Also, for data in Figure 5 and its supplements, can more information be determined e.g. Figure 5—figure supplement 1 could include additional PRE data; Figure 5—figure supplement 2, consideration of residues 63, 64; Figure 5—figure supplement 4, why is the back calculation for hexamer model A done for PRE data at 120 µM where hexamer will be minimally formed? The agreement is stated to be good, but this is not clear.

As requested, we have added additional raw PRE data in Figure 5—figure supplement 1 (new panel D).

Regarding residues 63 and 64 – we could determine their R_ex_ value at 180 μΜ, but not at 480 μΜ, as these resonances are too broad to measure at the higher concentration (hence the absence of a black bar). We have now amended Figure 5—figure supplement 2 (black crosses) to highlight residues that behave in this manner.

Figure 5—figure supplement 4 – the PREs measured at high protein concentrations suffer from line broadening due to the higher oligomer population and therefore several resonances disappear from the spectrum (red crosses in Figure 5—figure supplement 1). Thus, we chose to compare the models generated to the PRE data measured at the lower concentration where all three species contribute to the PRE data and the maximum number of resonances is observed.

6) Figure 5—figure supplement 5: Based on all 3 charge states, multiple hexamer models appear possible, however, the 15^+^ state is the main one considered. Please clarify the rationale for this analysis.

We discuss data only for the 15^+^ charge state, as this is the lowest charge state observed and hence the most compact and native-like. For the higher charge states (16^+^ and 17^+^) coulombic repulsion can result in partial or more complete protein unfolding. The consensus in the field (Vahidi et al., 2013) is that it is best to focus analysis on the lowest charge state observed, as this likely better reflects the solution phase structure of globular proteins using ESI-IMS-MS. We have edited the subsections “Structural models of on-pathway hexamers” and “Native ESI-IMS-MS”.

The Materials and methods describe ANS measurements for dimer, but no dimer data are shown. While not essential, it would be of interest to include such data for the dimer, and this may provide support for the modelled hydrophobic surface in the dimer. Please clarify the extent of fluorescence change when ANS binds to the Im7. The presentation is a bit confusing because hexamers show a larger increase ANS than monomer, but hexamers are stated to show only a small increase.

We have now added the ANS data for the dimer to Figure 5—figure supplement 5D. The dimers and hexamers each show a small increase in ANS fluorescence compared with monomers. However, given that the ANS intensity depends both on the amount of dye bound and the quantum yield of the dye in the bound state, we cannot quantitatively compare the differences in intensity with the structural models.

We chose to compare the ANS fluorescence in the presence of the ΔN6 oligomers and the folding intermediate of Im7, since the on-pathway Im7 folding intermediate is highly structured, reminiscent perhaps of the dimers and hexamers of ΔN6. Notably, the ANS fluorescence of the Im7 folding intermediate is much smaller than that observed when ANS binds to a typical ‘molten globule’ state, such the classical molten globule of apo α-lactalbumin (for which an ~100-fold increase in ANS fluorescence is observed (Semisotnov et al., 1991)). We have clarified the text in the subsection “Structural models of on-pathway hexamers” to hopefully make this point clear.

Dynamics:1) Please address when ΔN6 samples may or may not be at equilibrium in the various experiments, and how this may impact the data analysis.

Based on the CPMG data and the calculated exchange rates of 1790 s^-1^ and 205 s^-1^, equilibrium should be reached within 5 ms. Hence, all NMR experiments were performed under equilibrium conditions.

2) Subsection “A unified model of ∆N6 polymerization”: how does Figure 2B shows disappearance of hexamers during the elongation phase?

Figure 2B shows the appearance of a hexamer band at early stages of fibril formation that decreases in intensity as fibril formation proceeds. Since we did not carry out a simultaneous measurement of ThT fluorescence in this assay we cannot ascribe the% hexamer to a precise stage in the elongation kinetics. We have changed the text in the subsection “Native-like dimers and hexamers form during ΔN6 assembly” to reflect this fact more clearly, and we have added an electron micrograph to show that fibrils indeed form at the end of the reaction under the conditions employed (see revised Figure 2B).

3) It appears that relatively little monomer is consumed in the aggregation reactions (for example, consider the gel in Figure 2B). This is not addressed directly in the paper and it seems like having the majority of the material not aggregation prone must have a large effect on the interpretation of the data, i.e. the aggregation assays are reporting not on the major process that is occurring in the solution, but a relatively minor side reaction. There is not a gel for the pH 2 aggregation experiments, but it seems likely that must more of the monomer material is consumed?

The reviewers are correct in their observation that in seeded reactions at pH 6.2 not all of the initial monomer is consumed into fibrils, as shown in Figure 7C. Indeed, this is unusual behaviour compared with ‘typical’ seeding reactions where the elongating unit is a monomeric unfolded protein and the fibril yield is typically >90%. This is indeed observed for unfolded β_2_m at pH 2.0, where the critical concentration is <8 μΜ and reactions measured at concentrations of e.g. 100 μΜ have fibril yields of > 92%. However, ΔΝ6 at pH 6.2 elongates through a very different mechanism involving structured hexamers, whose population is highly dependent on the monomer concentration. After the hexamers have added onto the fibril ends and, due to the relatively high K_d_ for dimerization (~50 μΜ), the remaining monomer concentration rapidly becomes too low to enable dimer formation (and therefore hexamer formation), stalling the seeding reaction. As an illustrative example, at 500 μΜ ΔΝ6 where the population of ΔN6 molecules in hexamers is 56%, ~85 μΜ monomers (plus some dimers) remain in solution after fibril elongation – a concentration similar to the dimer K_d_. As seen in Figure 7C when the initial monomer concentration approaches the dimer K_d_ we effectively see no elongation. Therefore, the low fibril yield in the seeding reactions is a predictable outcome of the monomer-dimer-hexamer model and provides an independent validation of the notion that hexamer formation is the rate limiting step of fibril elongation.

4) Hexamer population is reported to correlate with amount of aggregated protein; please elaborate on the possible basis for this correlation.

This question has been answered in point 3 above.

Cell Toxicity:The cell toxicity studies are weak and should be removed; a single concentration at a single timepoint in a neuroblastoma cell line is that is of questionable physiological relevance is considered

One of the interesting findings of our manuscript is that the on pathway β_2_m dimers and hexamers do not exhibit cytotoxicity, at least under the conditions explored, and this was highlighted as a ‘noteworthy’ discovery in the editor’s summary of our paper. We prefer, therefore, to retain this information in our manuscript, as any reader would be bound to wonder why cytotoxicity was not assayed when amyloid oligomers are being discussed. However, we agree that our experimental observations are not exhaustive, and that we focus on a single cell type at a single protein concentration and incubation time, which we now discuss specifically in our revised manuscript (subsection “Structural models of on-pathway hexamers”).

The neuroblastoma cell line SH-SY5Y was chosen for our assays, as this cell line is a widely accepted model for the study of amyloid toxicity and has been used by other labs for β_2_m and other amyloid forming sequences. The choice of SH-SY5Y cells was also made since it enables a direct comparison with other published studies investigating oligomer-associated toxicity. A statement to this affect has been added to the manuscript and references are cited to justify this statement (see the aforementioned subsection).

Whilst only a single protein concentration was used for our assays, no effect on any of the assays for cell viability was observed, despite incubating the cells with a monomer equivalent concentration of ~3 μM cross-linked oligomers (calculated from the A_280_ of the hexamer fraction from the SEC column and the extinction co-efficient of monomeric β_2_m). Notably, the monomer equivalent concentration of the β_2_m hexamers is ~10 fold higher than that used by Fusco et al. (2017) to demonstrate that α-synuclein B* oligomers are cytotoxic.

A single time point was used, but again this corresponds to the timescales typically used in the references cited to determine whether oligomers are toxic. Whilst this allows comparison with other published studies, we accept that we cannot rule out toxicity taking longer to manifest. This caveat is now clearly stated in the manuscript.

[Editors' note: further revisions were requested prior to acceptance, as described below.]The manuscript has been improved but there are some remaining issues that need to be addressed before acceptance, as outlined below. Note that no further experimentation is required to address these issues.1) Questions regarding quantitative analysis remain:

*a) The proportions of monomer, dimer and hexamer for 120 μM total monomer are not consistent with those at higher concentrations (for 120 μM, the K_d_ dimer is 57 μM versus 50 μM, this should be addressed). The additional information on how the relative populations and K_d_s were calculated is pertinent, but not clear. Please define* konover *in the paper, give information on the origin of this equation (including reference), and include equations for how k_on_ overall is related to _τ_c and d.*

We are not sure what the reviewer is referring to regarding the dimer K_d_ at 120 μΜ. The values used throughout the paper are K_d, dim_ = 50 μΜ and K_d,hex_ = 10 x 10^-9^ M^2^. We checked and could find no mention of 57 _µ_M in our manuscript.

About konover (the overall rate of protein assembly, now defined in the subsection “Chemical shift perturbation and calculation of K_d_ values”) let’s consider a simple 3-state monomer (M), dimer (D), hexamer (H) model:

M→k1app←k-1D→k2app←k-2H

where k_1_^app^, k_2_^app^ are the apparent forward rate constants for dimer and hexamer formation respectively, k_-1_, k_-2_ are the reverse rate constants for dimerization and hexamerization, respectively. Based on elementary chemical kinetics the time evolution of the concentration of H and D is given by:

dH/dt=k2appD-k-2H (Equation 1)

dD/dt=k1appM-k1D-k2appD+k-2[H] (Equation 2)

In order to derive an expression for (konover) we simply need to eliminate [D] from Equation 1 and replace it with [M]. If we apply the equilibrium condition for Equation 2 (*d[D]/dt* = 0) we obtain:

D=k1appM+k-2[H]k-1+k-2app (Equation 3)

By substituting Equation 3 into Equation 1 and rearranging:

dH/dt=k1appk2appk-1+k2appM-k-1k-2k-1+k2app[H] (Equation 4)

with the term in front of [M] representing k^over^_on_. However, the reaction considered is not first order and we need to convert k_1_^app^, k_2_^app^ to rate constants using the relationships:

k1app=2k1Meq

k2app=3k2[Deq]2

Where, [M_eq_], [D_eq_] are the equilibrium concentrations of monomers and dimers. Therefore,

konover=k1appk2appk-1+k2app=6k1k2Meq[Deq]2k-1+3k2[Deq]2 (Equation 5)

The overall observed _τ_c and d depend on the shape of monomers, dimers and hexamers but most importantly on their populations. In turn, the oligomer populations depend on the ratio of k_1_, k_2_ with k-_1_, k-_2_ and the concentrations of monomers and dimers and hexamers at equilibrium. Since the off rates are not concentration-dependent, for simulation purposes we can fix their values, while restraining k_1_, k_2_ accordingly to satisfy the measured monomer and hexamer K_d_s. By doing so, the konovergiven by Equation 5 scales in the same way as the overall population of dimers and hexamers at any given monomer concentration and therefore _τ_c and d are expected to scale in the same way. [M_eq_], [D_eq_] were calculated by numerical integration of the monomer-dimer-hexamer model at τ=∞ using various initial monomer concentrations (M_tot_) and K_d, dim_ = 50 μΜ and K_d,hex_ = 10 x 10^-9^ M^2^. This discussion has now been added into the Materials and methods subsection “Chemical shift perturbation and calculation of K_d_ values”.

b) Some additional support is needed for the authors' conclusions regarding larger species not really being quantitatively observable for AUC, but also SEC experiments, where the absorbance for larger species appears lower than expected. Can the authors use absorbance values more quantitatively?

The fact that AUC at concentrations over 200 μΜ does not show hexamers is presumably because at these concentrations 20% of the material has already sedimented during the initial 3000 rpm run, consistent with the hexamers rapidly forming high molecular weight species that sediment before the c(S) data are acquired (Author response image 4).

**Author response image 4. respfig4:** Raw AUC data. Plot of absorbance at 280 nm across the AUC cell radius for the initial run (3000 rpm, blue) versus the first scan of the actual run at 48000 rpm (red).

One therefore would expect to be able to see high order species in the SEC of uncross-linked material. However, with the columns and sample loops used in the SEC experiment and using our estimated K_d_ values, a sample of 400 μΜ (hexamer population 48%) would be diluted roughly 5-times, resulting in a maximum hexamer population of only 6%. Thus, it is not surprising that we only observe monomers and dimers in the SEC of uncross-linked ΔΝ6. On the other hand, when crosslinked material is added on the column at high concentrations (400-500 μΜ) a peak is observed at the void volume even at 0h (black curve Figure 2—figure supplement 1D) consistent with rapid formation of larger species, consistent with the AUC data. These complications do not allow detailed quantification of AUC/SEC absorbance values, other than identification of the MW of the species present. This discussion has now been added to the subsection “Chemical cross-linking and analytical SEC”.

c) Additional information is needed to show that independently prepared samples may indeed be compared i.e. they do not differ in uncontrolled ways due to unaccounted for variations in the timecourses or levels of assembly (oligomerization/aggregation), i.e. can the samples be considered to be consistently comparable (prior to and after completion of the time course of assembly/aggregation)?

More than 20 NMR samples from at least 10 different protein preparations over 4 years were used in the present study. All of those gave consistent results, while controls (i.e. HSQCs) were routinely carried out in order to check sample quality before and after each NMR experiment.

d) Figure 2B: hexamer appears at 48 hours and disappears, while monomer appears unchanged. Presumably this is because at 80 μM total monomer, relatively little hexamer forms and gives rise to aggregates. So, one may reach an equilibrium at ~100 hours, where only monomer and dimer are significantly populated. It may be helpful to add some text (similar to response to Dynamics point 3) or a figure to make this point clear, earlier in the paper. As for the lack of observation of dimer, could this be because the dimer is not effectively crosslinkable?

The reviewer is right in pointing out that at 80 μΜ the population of hexamers is only 6%. The apparent appearance and disappearance of the hexamer band over time in Figure 2B shows that the hexamers go on to form aggregates that do not enter the gel. As for the absence of a dimer band, based on our dimer model only one lysine pair between residues 58 in monomer 1 and 91 in monomer 2 can potentially be crosslinked, making the dimer less amenable to crosslinking in comparison to the hexamer where many different crosslinks could be formed. We now note this in the figure legend.

e) Given the experimental considerations the normalization of the chromatography signals may be reasonable; however, the relative signals within each panel are not internally consistent. Why is this?

The main point of the SEC experiments is to show that in the absence of cross-linking we only observe monomers and dimers while after cross-linking an array of oligomers are observed, with dimers and hexamers being the most prevalent ones. Data in Figure 2—figure supplement 1B show a decreased monomer population at higher protein concentrations fully consistent with hexamer species becoming more prevalent. For the uncrosslinked samples in Figure 2—figure supplement 1A, the dimer peak increases as the protein concentration is increased as expected. The unavoidable dilution of the samples and their re-equilibration during the SEC run, and also the complications discussed in point b above, make it impossible to quantitate the relative peak intensity in SEC traces of the uncrosslinked samples. However, this does not affect our conclusion from these data, which was simply to show that crosslinking increases the number of oligomers observed. This discussion has now been added to the subsection “Chemical cross-linking and analytical SEC”.

2) Questions remain regarding the toxicity assay:a) The toxicity assays are at best only a weakly supporting data: it is unclear why this is at all relevant for β_2_-microglobulin which does not specifically impact neuronal cells. It seems likely the cells simply don't take up β_2_-microglobulin, and good positive controls for these experiments are lacking.

SH-SY5Y cells are a commonly used model for amyloid toxicity and were chosen since they allow comparison of the toxicity of the β_2_m hexamers with studies of the toxicity of other oligomers in amyloid formation. The observed lack of toxicity of the dimers and hexamers is unlikely to be due to a failure of these species to be taken up by the cells. Indeed, we have shown previously that SH-SY5Y cells not only take up the β_2_m monomers, but also β_2_m amyloid fibrils (Jakhria et al., 2014). This is now stated in the text (subsection “Structural models of on-pathway hexamers”).

The legend for Figure 5—figure supplement 6 has been amended to include additional information on the assay controls and how the data are normalised. For MTT reduction, ATP levels and ROS production, the data are normalized to the PBS buffer control (100%) and NaN_3_ treated controls (a positive control for cell death (0% )). LDH release is normalized to detergent lysed cells (a positive control for cell lysis, 100% ) and PBS buffer treated controls (0% ). Additional text has also been added to the Materials and methods (subsection “Cytotoxicity assays) describing H_2_O_2_ as a positive control for the induction of ROS. We have placed the raw data for these in the University of Leeds data repository so they are accessible to interested readers.